



María Barrera-Verdejo (mbarrera@smail.uni-koeln.de)
Manuscript prepared for J. Name
with version 2014/09/16 7.15 Copernicus papers of the LaTeX class copernicus.cls.
Date: 22 February 2016

# Ground Based Lidar and Microwave Radiometry Synergy for High Vertical Resolution Absolute Humidity Profiling.

María Barrera-Verdejo[1], Susanne Crewell[1], Ulrich Löhnert[1], Emiliano Orlandi[1], and Paolo Di Girolamo[2]

[1]Institut für Geophysik und Meteorologie, Universität zu Köln
[2]Scuola di Ingegneria, Universita degli Studi della Basilicata

*Correspondence to:* María Barrera-Verdejo (mbarrera@smail.uni-koeln.de)

**Abstract.** Continuous monitoring of atmospheric humidity profiles is important for many applications, e.g. assessment of atmospheric stability and cloud formation. Nowadays there is a wide variety of ground-based sensors for atmospheric humidity profiling. Unfortunately there is no single instru-

ment able to provide a measurement with complete vertical coverage, high vertical and temporal resolution, and good performance under all weather conditions, simultaneously. For example, Raman lidar (RL) measurements can provide water vapor with a high vertical resolution albeit with limited vertical coverage, due to sunlight contamination and the presence of clouds. Microwave radiometers (MWR) receive water vapor information throughout the troposphere though their vertical resolu-

tion is poor. In this work, we present a MWR and RL system synergy, which aims to overcome the specific sensor limitations. The retrieval algorithm combining these two instruments is an Optimal Estimation Method (OEM), which allows for an uncertainty analysis of the retrieved profiles. The OEM combines measurements and a priori information taking the uncertainty of both into account. The measurement vector consists of a set of MWR brightness temperatures and RL water vapor pro-

files. The method is applied to a two month field campaign around Jülich (Germany), focusing on clear sky periods. Different experiments are performed to analyse the improvements achieved via the synergy compared to the individual retrievals. When applying the combined retrieval, on average the theoretically determined absolute humidity uncertainty can be reduced by 60% (38%) with respect to the retrieval using only-MWR (only-RL) data. The analysis in terms of degrees of freedom per

signal reveals that most information is gained above the usable lidar range, especially important during daytime when the lidar vertical coverage is limited. The retrieved profiles are further evaluated using radiosounding and GPS water vapor measurements. In general, the benefit of the sensor com-



bination is especially strong in regions where Raman lidar data is not available (i.e. blind region, regions characterized by low signal to noise ratio), whereas if both instruments are available, RL
dominates the retrieval. In the future, the method will be extended to cloudy conditions, when the impact of the MWR becomes stronger.

## 1 Introduction

Highly resolved, accurate and continuous measurements of water vapor are required for a deeper understanding of many atmospheric phenomena (Stevens and Bony, 2013). Specifically, processes on
short time scales such as convection, cloud formation or boundary layer turbulence are challenging due to their high associated water vapor variability, which is difficult to capture by one instrument alone (Steinke et al., 2014). In order to overcome this limitation, the scientific community started merging different data from several instruments in the last 15 years.

Some examples for ground-based synergies have been proposed by Stankov (1998), Löhnert et al.
(2004), Furumoto et al. (2003) or Bianco et al. (2005) and Delanoe and Hogan (2008) for satellite applications. In the present paper, the synergy between ground based Raman Lidar (RL) and Microwave Radiometer (MWR) is described. Both instruments present some advantages and disadvantages and, by bringing them together in an optimal and new retrieval algorithm, it is possible to overcome some of the disadvantages of the single devices and enhance their benefits.

Water vapor RL systems provide humidity profiles with high vertical resolution. For this reason, such lidars became a powerful tool in active ground based observations during the last years. New retrieval algorithms optimally exploiting the information content have been developed (Sica and Haefele, 2015; Povey et al., 2014). However, the RL techniques alone show some drawbacks, which hinder the operational application. For example, ground based RL cannot provide information above
and within optically thick clouds, as the radiation emitted by the lidar is severely attenuated once the laser beam reaches a liquid layer within the cloud. Moreover, day time measurements are affected by solar background radiation, which strongly reduces the vertical extent of the profile. The continuous and effective detection of the weak Raman signals demands a robust and stable alignment of the receiving system. Daytime operation requires the use of powerful lasers whose continuous operation is
technically demanding. Additionally, RL needs to be regularly calibrated.This calibration is usually performed based on the use of radiosounding data, which presents some caveats. First, the balloon might measure a different air volume due to its drift. Second, it implies rather high costs, both instrumental as well as human resources. The calibration of the lidar is a key point that still stimulates new solutions (Foth et al., 2015). In addition, lidar data from the lowest atmospheric layers typically
cannot be used, due to the presence of a blind region (or zero overlap region (ZOR)) associated with the overlap function (OVF) of the RL.



The MWR allows automated continuous data acquisition and is a robust operational instrument (Rose et al., 2005), measuring unattended in a 24/7 mode. Brightness temperature measurements at different features allow the determination of humidity and temperature profiles. In contrast to RL, the instrument offers a limited vertical resolution in the retrieved atmospheric profiles, especially in the higher layers of the atmosphere (i.e. above an altitude of 1km) (Löhnert et al., 2007), but performs best for measurements close to the ground, where there lidar data are missing. MWR also provides accurate integrated quantities such as Integrated Water Vapor (IWV) or Liquid Water Path (LWP) (Crewell and Löhnert, 2003; Löhnert and Crewell, 2003). The calibration of this instrument is performed with internal and external references with known temperature (hot load-cold load) or by observing the atmosphere under different elevation angles (i.e. sky tipping) (Maschwitz et al., 2013). An advantage of the MWR is its capability of measuring in almost all weather conditions (also cloudy cases) except for rainy scenarios, where the received signal must be discarded in most of the cases.

A method to combine RL and MWR was already proposed by Han et al. (1997), where the authors developed a two-stage algorithm to derive atmospheric water vapor profiles. In the first stage, a Kalman filtering algorithm was applied using surface in situ and RL measurements. In the second stage, a statistical inversion technique was applied to combine the Kalman retrieval (used as prior information, not as observations) with the integrated water vapor of a two-channel MWR and climatological data. Their method showed that the synergy of these two sensors compensates for the individual sensor's drawbacks. A continuation of this work was carried out by Schneebeli (2009) who, still following the Kalman filter two-stage configuration, extends this approach to also temperature profiles.

The method described in this document is a new approach based on an Optimal Estimation Method (OEM), an iterative optimal and physically consistent method that allows uncertainty assessment and provides the most probable estimated atmospheric state together with its uncertainty description. The aim of this study is to combine the information provided by the two instruments in an OEM to retrieve atmospheric water vapor profiles. Note that this flexible framework allows to incorporate the retrieval of temperature once corresponding RL and MWR data are available. The method was applied to the 2-months dataset collected during HOPE ($HD(CP)^2$ Observational Prototype Experiment), where a multitude of ground-based remote sensing instruments for the investigation of boundary layer and cloud processes were operated (Steinke et al., 2014; Behrendt et al., 2015; Foth et al., 2015). Here we focus on clear sky cases and absolute humidity (AH) profiles. A description of the method is presented in section 3. Section 4 describes the results when the OEM is applied to a case study, whereas Section 5 evaluates the OEM when applied to the two months period of HOPE. Finally, section 6 summarizes the results and provides an outlook.



## 2 Observations: HOPE

In this study we make use of the data collected during HOPE (HD(CP)2 Observational Prototype Experiment) , which was a major field campaign in Nordrhein-Westfalen, Germany, from April to June
2013. One main goal of HOPE was to provide information on subgrid variability (i.e. of water vapor) and microphysical properties on scales smaller than 1 km, which corresponds to the horizontal resolution of state-of-the-art operational mesoscale models. During the measurement period, three supersites were operating that were distributed within the 5-10 km surroundings of Forschungszentrum Jülich, Germany (50.905, 6.411944). Each supersite was composed of a rich variety of remote sensing instruments such as cloud radar, lidar and microwave radiometer. A large set of of more than 200 radiosondes (RS) was launched only 4 km away from the JOYCE (Jülich ObservatorY for Cloud Evolution) site and at least twice a day.

At the permanent supersite JOYCE (Löhnert et al., 2014), measurements by the University of Basilicata Raman Lidar system (BASIL) and a MWR were carried out and auxiliary data from other instruments is available.

### 2.1 BASIL

The Raman Lidar system BASIL (Di Girolamo et al., 2009; Di Girolamo et al., 2012) is an active instrument detecting the elastic and Raman backscattered radiation from atmospheric constituents. BASIL includes a Nd:YAG laser emitting pulses at its fundamental wavelength, its second and third harmonics: 355, 532 and 1064 nm, respectively, at a repetition rate of 20 Hz. For the purpose of water vapor profiling, Raman scattering stimulated by the 355 nm beam is used because of the higher cross-section with respect to other wavelengths. The average power emitted at 355 nm is 10 W. Nevertheless, other transmitting wavelengths could also be used for water vapor detection, as reported by Althausen et al. (2000). The receiver is built around a larger telescope in Newtonian configuration (45 cm diameter primary mirror) and two smaller telescopes (5 mm diameter lenses). The larger telescope is primarily dedicated to the collection of the Raman signals, i.e. the water vapor and molecular nitrogen roto-vibrational Raman signals, at 407.5 and 386.7 nm, respectively, which are used to estimate the water vapor mixing ratio profiles.

Signal selection is performed by means of narrowband interference filters, whose specifications were reported in Di Girolamo et al. (2004) and Di Girolamo et al. (2009). Sampling of the Raman signals is performed by means of transient recorders with double signal acquisition mode (i.e. both analog, A/D conversion and digital, photon counting). Depending on the application, water vapor mixing ratio profiles can be derived with different vertical and temporal resolutions. These two parameters can be traded-off to improve measurement precision. For the purposes of this study, the lidar products are characterized by a vertical resolution of 30 m and a temporal resolution of 5 minutes. Because of the absence of overlap between the laser beam and receiver field-of-view,



there is a blind region in the lower altitudes. Consequently, vertical profiles of water vapor mixing ratio typically start at 150-180 m above ground. Humidity profiles extend vertically up to different altitudes during daytime and nighttime depending on the altitude where the signal gets completely

extinguished. For water vapor, considering a vertical/temporal resolution of 30 m/5 minutes, this typically takes place around 4-5 km during daytime and around 12 km during the night. The different ranges result from the additional noise due to solar contamination during daytime.

During HOPE, BASIL has been calibrated based on the comparison with the radiosondes launched approximately 4 km away from the instrument. A mean calibration coefficient was estimated com-

paring BASIL and radiosonde data. Every clear sky radiosonde coincident with BASIL measurements (sixty in total) is compared to the lidar profile in an altitude region with an extent of 1 km above the boundary layer. We choose this region to minimize the air mass differences related to the distance between the lidar station and the radiosonde launch facility station. For every profile comparison, a value for the calibration constant is calculated. Out of these 60 values, we calculate the

mean value and use it as the calibration constant for the complete period of HOPE. The standard deviation of the mean calibration coefficient from the single values does not exceed 5%.

In addition to the calibration constant uncertainty, other smaller systematic uncertainty sources might affect the water vapor measurements. For example, an additional uncertainty (<1%) may be considered related to the use of narrowband filters, the temperature dependence of $H_2O$ Raman

scattering and the thermal drifts of the filters (Whiteman, 2003). Further, an additional 1% may be associated with the determination of the differential transmission term at the water vapor and molecular nitrogen Raman wavelengths (Whiteman, 2003). These sources of uncertainty, in principle negligible, are not taken into account for the calculations in our algorithm.

The statistical uncertainty of the water vapor mixing ratio is calculated based on the application

of the Poisson statistics (Di Girolamo et al., 2004) and varies for each range bin. Providing a profile with 5 minutes time-resolution and 30 meters vertical grid, the statistical uncertainty affecting water vapor mixing ratio measurements for night-time operation is typically smaller than 2% up to 3 km and smaller than 20% up to 9 km. For daytime operation, it is typically smaller than 40% up to 3 km and smaller than 100% up to 4.5 km.

Note, the operation of BASIL has not been continuous during HOPE. The instrument collected a total of 430 hours of measurements distributed over 44 days, which represents 30% of the whole HOPE period.

## 2.2  MWR

The microwave radiometer profiler HATPRO (Rose et al., 2005) was manufactured by Radiome-

ter Physics GmbH, Germany (RPG) as a network-suitable microwave radiometer allowing retrieval of Liquid Water Path (LWP) and Integrated Water Vapor (IWV) at high temporal resolution (1 s) (Crewell and Löhnert, 2003). It is a passive MWR that measures radiation expressed as brightness



temperature in two frequency bands (Rose et al., 2005). The seven channels of the K band contain
information about the vertical profile of humidity through the pressure broadening of the optically
thin 22.235-GHz $H_2O$ line. This band also provides the information for determining LWP as emis-
sion by liquid water is increasing with frequency. The seven channels of V-band contain information
on the vertical profile of temperature resulting from the homogeneous mixing of $O_2$ throughout
the atmosphere (Löhnert and Maier, 2012). For temperature, retrieval improvement can be obtained
by including off-zenith observations under the assumption of horizontal homogeneity, however for
water vapor profiling, only zenith observations are beneficial (Löhnert et al., 2009).

The absolute calibration of the instrument is performed roughly every six months, taking a cold
and a hot load as references, which are assumed to be ideal black bodies. The cold black body is
a liquid-nitrogen-cooled load at approximately 77 K that is attached externally to the radiometer
box. This standard, together with an internal ambient black body load inside the radiometer, is used
for the absolute calibration procedure (Maschwitz et al., 2013). In addition, a calibration by tip-
curve observations can be performed for the K-band channels using obervations at different elevation
angles (Turner et al., 2007). The reliability of sky tipping calibrations strongly depends on how good
the assumption of a horizontally stratified atmosphere is. Further details on the calibration procedures
of the instrument can be found in Maschwitz et al. (2013).

The temporal resolution of this instrument is higher than for the RL: it is able to provide one mea-
surement every 1-3 seconds. Thus, a temporal adaptation to the lidar time resolution is performed,
averaging MWR measurements in five minutes intervals. A major drawback of MWR water vapor
and temperature profile retrievals is the limited vertical resolution. Typically, only two pieces of in-
dependent information for water vapor profiles are contained in the measurements, whereby 3-4 are
obtained for the temperature profile (Löhnert et al., 2009).

## 3   Method

### 3.1   Optimal Estimation Method

An Optimal Estimation Method is applied which allows estimating the state of the atmosphere and
its associated uncertainty. Using this scheme requires a set of measurements (with their uncertainty
specification), a forward model, which relates the atmospheric state to the instrument measurements,
and some *a priori* information. In the following, a short description of the scheme is presented. More
details can be found in Rodgers (2000).

Given the *moderately non-linear* nature (Rodgers, 2000) of our problem, the iterative equation
applied to find the best atmospheric state estimate is:

$$x_{i+1} = x_a + (S_a K_i^T (K_i S_a K_i^T + S_\epsilon)^{-1} [y - F(x_i) + K_i(x_i - x_a)]) \tag{1}$$




where $x_i$ is a vector containing the atmospheric state at the iteration $i$. The observation vector $y$ contains the brightness temperatures (TB) from the MWR and the mixing ratio from the lidar. The term $x_a$ represents the *a priori* information of the atmosphere, in our case, coming from radioson-des. $S_a$ and $S_\epsilon$ are the covariance matrices of the prior and observation uncertainties, respectively.

$F(x_i, b)$ is the forward model applied to the state vector $x_i$, and depending on the model parameters $b$. For simplicity, it will be referred as $F(x_i)$ in the following. The forward model output lies in the observation space. The term $K$ represents the Jacobian, which can be understood as the response the observation vector when a perturbation is subject to the atmospheric state vector (eq. (2)):

$$K_i = \frac{\partial F(x_i)}{\partial x_i} \tag{2}$$

The iterative equation described in (1) finds the most optimal atmospheric state $x_{op}$. Convergence of the solution is reached once the convergence criterium is fulfilled, i.e. the difference between the observation estimates at iterations *n* and *n+1* is one order of magnitude smaller than the estimated error. To evaluate this difference we must scale the change in the solution by its estimated error, leading to:

$$d_i^2 = (F(x_{i+1}) - F(x_i))^T (S_\epsilon (KS_aK^T + S_\epsilon)S_\epsilon)^{-1}(F(x_{i+1}) - F(x_i)) < m/10 \tag{3}$$

where m is the number of elements in the observation vector. An uncertainty estimation of the solution $S_{op}$ is calculated via:

$$S_{op} = S_a - S_aK^T(S_\epsilon + KS_aK^T)^{-1}KS_a \tag{4}$$

where K is the Jacobian calculated in the last iteration. It is also possible to estimate the infor-

mation content of the result. The degrees of freedom (DOF) of a profile represent the amount of independent pieces of information in the signal. They can be calculated as the trace of the matrix in the following equation (5):

$$A_{ker} = S_aK^T(S_\epsilon + KS_aK^T)^{-1}K \tag{5}$$

where $A_{ker}$ is the averaging kernel. This matrix is very important to describe the information

content, as it describes the subspace of the *state space* in which the retrieval must lie. Its diagonal elements can be seen as a measure of the number of degrees of freedom per discrete altitude level. The reciprocal denotes the number of levels per degree of freedom and is can be interpreted as a



measure of resolution. The vertical resolution $\Delta z$ is thus defined as the range of heights covered divided by the number of independent quantities measured:

$$\Delta z = \frac{\delta z}{diag(A_{ker})} \tag{6}$$

where $\delta z$ is the vertical spacing grid for the retrieval. It is important to note the difference between the vertical discretization of the retrieved profile and the quantification of the *effective vertical resolution* $\Delta z$.

### 3.2 A priori: $x_a$ and $S_a$

The a priori information is calculated from the set of radiosondes launched during HOPE. A total of 217 sondes have been considered. Generally, at least two of them are available for every day of the campaign, typically one around noon and the other at midnight. From these data, the average profile of absolute humidity $q$, in $kg/m^3$, is calculated to represent the a priori knowledge, together with its standard deviation $s_q$. This profile is used as $x_a$ in the algorithm described by eq. (1).

For the same set of radiosondes, the correlation (*corr*) and covariance (*cov*) matrices are calculated according to Wilks (2006), to describe the relation of absolute humidity between two different altitude levels. We can define $q$ that represents absolute humidity as a function of the altitude:

$$q = [q_1, q_2, ... q_k] \tag{7}$$

being $k$ the total number of altitudes in the retrieval vertical grid. Therefore, the *corr* and *cov* matrices have a dimension of $k \times k$, calculated with the formula:

$$corr_{q_a,q_b} = \frac{cov(q_a,q_b)}{s_{q_a} s_{q_b}} = \frac{\frac{1}{n-1}\sum_{i=1}^{n}[(q_{a_i} - \bar{q_a})(q_{b_i} - \bar{q_b})]}{\left[\frac{1}{n-1}\sum_{i=1}^{n}(q_{a_i} - \bar{q_a})^2\right]^{\frac{1}{2}}\left[\frac{1}{n-1}\sum_{i=1}^{n}(q_{b_i} - \bar{q_b})^2\right]^{\frac{1}{2}}} \tag{8}$$

where $i$ denotes each radiosonde, with a total of $n = 217$. The parameter $\bar{q}$ is the mean vertical profile of absolute humidity, and $a$ and $b$ are indices for all the different $k$ altitudes.

Both covariance and correlation matrices have been calculated as in equation (8). The first is needed in the retrieval algorithm as input ($S_a$), the second because it better illustrates the relations between water vapor at different altitudes in the atmosphere. The correlation matrix (Fig. 1) illustrates how the absolute humidity at a certain altitude is correlated with the one at other altitudes, from ground to 10 km. The values for the correlation are strongest close to the main diagonal, but decrease quickly for off diagonal terms. In the lowest 1-2 km there is a higher correlation, because of the well mixed conditions in the boundary layer. The result is similar to previous studies (Ebell et al., 2013).




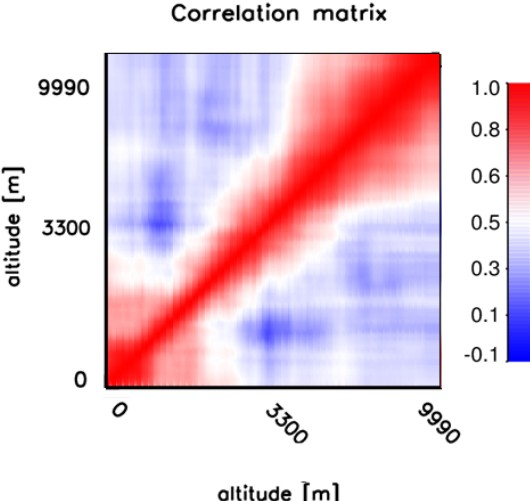

**Figure 1.** Correlation matrix derived from 217 radiosondes launched during HOPE. Correlation is shown for absolute humidity as a function of the altitude (from 0 to 10 km above the ground).

### 3.3 Observations: $y$ and $S_\epsilon$

The measurement vector $y$ is composed of the TBs from the MWR and the water vapor mixing ratio (WVMR) profile from the RL. Its size is variable, since it depends on the number of values the

lidar is able to measure at every given time interval. A lidar mixing ratio profile [kg/kg], together with its statistical uncertainty, is provided with a vertical resolution of 30 m, starting from 180 m (See Sec. 2.1). Below this altitude, the lidar detectors cannot be interpreted in a meaningful way to due lack of overlap of the emitted and received beams. Due to decreasing signal to noise ratio with height, one must determine the altitude up to which the lidar data can be considered meaningful.

This altitude range has been defined via the relative uncertainty of the WVMR, which is calculated at each altitude as the ratio between the uncertainty and the measurement itself. When this value is larger than 100%, the data is considered too noisy and is discarded. Care is required when applying this threshold, because possible random peaks in the lidar uncertainty can to lead to a filtering of too many points. Therefore, a running average is performed on the data with a 300 m window size

in the vertical. This smoothed profile is only used to select the clipping altitude for the RL data. The 100% uncertainty altitude is reached at different heights depending on the weather situation or night/day-periods. Typically it was found around 3-4 km during daytime and around 7-8 km during nighttime measurements.

In effect, the observation vector $y$ is composed of $t + m$ elements, and $S_\epsilon$ is a matrix with dimen-

sions $(t + m, t + m)$; $m$ being the number of altitudes where the lidar measurements have sufficient


signal to noise ratio, and $t$ is the number of TBs. Seven brightness temperatures are used for the retrieval of absolute humidity. Note, that within the retrieval procedure TB from the MWR are used directly in the measurement vector, while an atmospheric state variable (WVMR) is used from the lidar to complete the measurement vector that only requires a conversion of humidity units.

The error covariance matrix associated with the MWR measurement (with dimensions $(7 \times 7)$, is obtained empirically by calculating the covariance for the different channels, while constantly viewing an ambient black-body target with known temperature. The diagonal elements represent the covariance of each channel with itself, typically with values around the noise level ($\sim 0.25$ K). The off-diagonal elements represent the covariance between the measurements of different channels.

Because the channels share some electronical components inside the instrument, the off-diagonal elements cannot be considered zero, but typically show values one order of magnitude smaller than the main diagonal.

    The part of $S_\epsilon$ corresponding to the RL (dimension $(m \times m)$ is defined as a diagonal matrix containing only the random uncertainty at every altitude. This definition implies no correlation between

measurements in different heights. This simplification in the error covariance matrix has been also considered by other authors (Wulfmeyer et al., 2006; Dunbar et al., 2014; Adam et al., 2015). The $S_\epsilon$ elements corresponding to the correlation between RL and MWR measurements have been set to zero because no correlation is expected among measurement uncertainty of two separate instruments.

### 3.4   Forward models (FM)

The forward model for the lidar is straightforward because in our retrieval approach we consider WVMR as part of the measurements vector. Therefore, the lidar FM for water vapor simply performs the conversion from absolute humidity to mixing ratio. The FM for the MWR involves a radiative transfer model (Löhnert et al., 2004). It considers emission and absorption of radiation by gases in the atmosphere but neglects scattering, which can be ignored for all atmospheric particles except for

rain droplets. The model divides the atmosphere in layers and calculates the optical thickness and absorption coefficients at each level. From these values, and applying the radiative transfer equation (9) (Janssen, 1993), the TBs are calculated:

$$TB_{ground} = TB_{cos} exp(-\tau) + \int\limits_0^\infty T(s)\alpha(s)exp(-\int\limits_0^s \alpha(s')ds')ds \tag{9}$$

    Where $\tau$ is the optical depth of the whole atmospheric column (opacity), $\alpha$ is the absorption

coefficient $[m^{-1}]$ and $TB_{cos}$ is the cosmic background radiation (approx. 2.7 K).

    The retrieval vertical grid is defined for every profile. It varies, as well as the observation vector, depending on the amount of available lidar information for every given profile. In the atmospheric regions where lidar data is available, the vertical grid of the retrieval product is 30 meters (same as





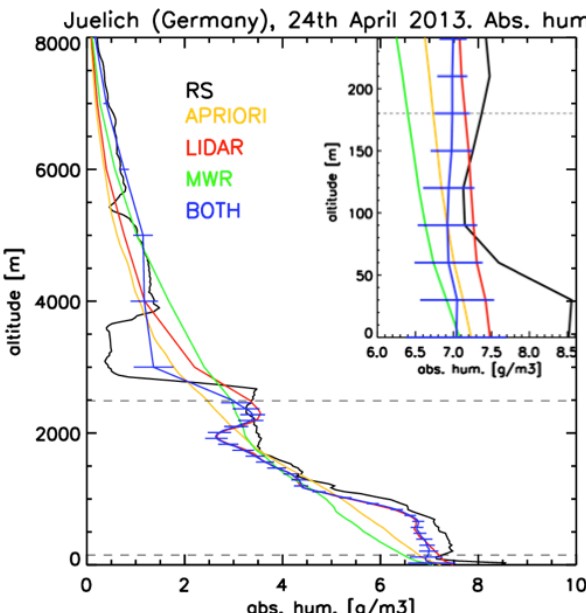

**Figure 2.** Absolute humidity profiles for a priori (yellow), only-RL (red), only-MWR (green) and MWR+RL (blue). The RS is used as reference (black). The dashed horizontal lines enclose the region where the lidar data is used. The inset is a zoom for the region close to the ground, between 0 and 250 m.

the lidar). Above the point where the RL signal is lost, and since the MWR cannot provide such high
resolution, the algorithm retrieves only one value every 1 km.

## 4 Application of the OEM: case study

### 4.1 Single profile retrieval

In a first approach, the OEM is implemented for the combination of the two instruments to retrieve
atmospheric absolute humidity. The setup is designed such that the OEM can work with input from a
single instrument as well. This aspect allows us to compare the performance of each sensor working
alone in contrast to the combination of the both. In the following, we demonstrate the algorithm
presenting the results corresponding to the 24th of April at 11 UTC, where a collocated RS is used
only as reference (Fig. 2). The a priori profile is the prior atmospheric knowledge, and also the
starting point (first guess) for the algorithm iteration.
At first, we introduce in the OEM only the portion of profile where RL data is considered to be
valid (i.e. from 180 m to 2.5 km, $\sim$ 77 layers), not taking into account the MWR yet. The result of
the algorithm is a complete profile from ground up to 10 km. In the region with lidar availability, the





result is strongly constrained by the lidar observations, since the associated uncertainties are very small (on the order of $0.5\ g/m^3$). In the regions with no lidar data, the profile is completed with the

information provided by the a priori profile and the a priori covariance matrix. Second, if only the seven TBs of the MWR are introduced in the OEM, a very smooth profile is obtained. This is because the seven frequencies do not provide enough information to distinguish fine vertical structures: MWR can only provide $\sim 2$ DOF per profile, as already mentioned in section 2.2. Therefore, the a priori profile plays a dominant role for defining the vertical structures. Finally, the output profile for

the RL and MWR combination is strongly constrained to the RL observations from 180 m to 2.5 km. Outside this region, the profile is completed based on the information provided by the TBs and the a priori.

The OEM uncertainty of the combined retrieval is calculated within the algorithm as well (eq. 4). The error is small in the region where there is RL data available ($\sim 0.5g/m^3$), but it increases

with altitude, as to be expected 2. It is also slightly larger close to the ground ($\sim 1g/m^3$), due to the absence of lidar data. Throughout the profile, the combined retrieval uncertainty is smaller than the only-RL and only-MWR ones. (refer also to Section 5.3 for detailed uncertainty statistics).

The profile obtained with the RL-MWR combination best fits the RS (shown as reference), launched at the same time 4 km away. Only the combined retrieval can detect the drop in humidity at 3 km and

the increase at 5 km. This is due to both the additional microwave radiometer observations as well as propagated lidar information (via the a priori covariance matrix). It is interesting to pay attention to the lower part of the atmosphere, close to the ground. In Fig. 2, a zoom from 0 to 250 meters is shown. Due to the missing RL information below 180 meters, the RL-MWR combination tends to the MWR values close to the ground, but quickly approaches to the lidar, as soon as the first RL

values are available. One can see that the lowest values of the RS are $1-1.5g/m^3$ more humid than the rest of the profiles. This might be explained by the fact that the sonde has been launched under different local environmental conditions: while the instruments site is located inside the research center, the RS is launched in an open field area. In addition, the venting of the RS is not optimal in the lowest 100 m. These could cause slight differences in the comparisons close to the ground, but

should not be a problem in the free troposphere.

We can additionally evaluate the quality of our retrieval by calculating the *effective* vertical resolution. Fig. 3 presents the vertical resolution $\Delta z$ calculated with eq. (6) for the three different retrievals on the 24th of April 2013, at 11 UTC. The results nicely show the improvements of the MWR+RL combination. In the region where RL is available (from 180m to 2.5 km), the only-RL resolution is

very high ($\sim 100-300m$). But outside this region, the vertical resolution for only-RL adopts the value of infinite. The only-MWR resolution is always coarser: up to 2.5 km it presents values one order of magnitude larger than the other two cases. Nevertheless, the advantage of the MWR is that the instrument provides information throughout the complete profile. Finally, the MWR+RL case presents the lowest vertical resolution. It adopts similar values as the only-RL resolution when RL is



available, and improves the resolution by $\sim 1 - 2km$ compared to the only-MWR case throughout the rest of the profile. Since the solution is strongly constrained by the lidar observations between 180 m and 2.5 km, the additional information contained in the MWR observations is now mainly distributed in the region above the 2.5 km.

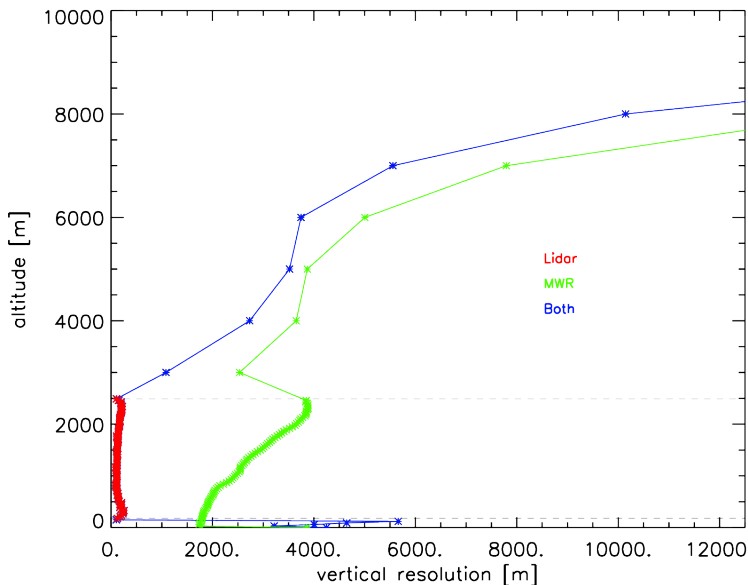

**Figure 3.** Vertical resolution for the only-RL (red), only-MWR (green) and MWR+RL (blue). The dashed lines enclose the area where RL data has been considered.

## 4.2 Time series

The same methodology is now applied to a larger measurement period. An example of this is shown in Figure 4, which presents a 7-hour time series of the absolute humidity on the 17th of April 2013 during HOPE. The figure shows the effect of a cold front passage around 23 UTC leading to an intrusion of dry air into the lower altitudes. Above 5 km, no significant changes of humidity occur. The time series reveals a successful synergy between RL and MWR, making use of the TB and a

priori information to complete the profile where RL measurements are not available (i.e. in the blind region below 180 m and at regions of too high a lidar noise level).





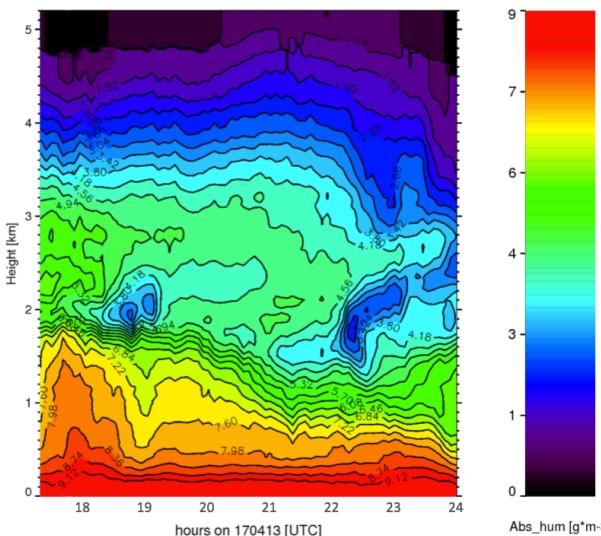

**Figure 4.** Time series of the joint retrieved absolute humidity, in the afternoon of the 17th April 2013.

## 5 Application of the OEM: Statistics over HOPE

The absolute humidity algorithm has been applied to all the clear sky periods with simultaneous availability of MWR and RL. The MWR measured continuously, so this selection is restricted to lidar availability. There are in total 4201 lidar profiles (30% of the total campaign). Out of them, 717 profiles are considered as clear sky (around 17% of the total). Out of all the clear sky profiles, the convergence of the OEM is found in 95.8% of the cases, that is, 687 profiles. In the rest of the cases, the convergence is not reached because the algorithm cannot find a profile which is simultaneously consistent with the measurements of the two instruments and the a priori, within their uncertainties.

### 5.1 Integrated Water Vapor

Another key atmospheric parameter that we can evaluate after applying the OEM is the IWV. The independent measurements of IWV from the Global Position Satellite (GPS) ground station (Bevis et al., 1992) can be used to assess the quality of the retrieval products. In Fig. 5(a), the time series of the IWV during HOPE is presented. The continuous IWV signal from GPS measurements is shown together with the IWV from the joint retrieval, which is only available during clear sky events. IWV reveals strong fluctuations with values between 5 and 29 $kg/m^2$ during HOPE, and therefor this period is well suited for evaluation studies.

Fig. 5(b) quantitatively compares the three OEM retrieval cases (combined retrieval, MWR and only-RL) against the GPS signal. Note, that a comparison with the original lidar data before processing in the OEM is not sensible, since the lidar lacks information in the lowest atmosphere (due to




incomplete overlap) and also above the altitude where the SNR is too large. A sensible comparison is only carried out after OEM processing because these retrieval results provide full profiles in all three cases.

Fig. 5(b) also shows the values for the bias and the standard deviation (in $kg/m^2$) for all the cases.

The values are small in all situations and lie inside the GPS uncertainty of $1 - 2kg/m^2$ (Gendt et al., 2004) and the MWR product of $\sim 0.5 - 1kg/m^2$ (Steinke et al., 2014). The combination of the two instruments and the only-MWR case present similar standard deviations, whereas the only-RL case presents a twice as large standard deviation in comparison to the other two cases. These numbers confirm that the RL cannot provide any additional information to the calculation of the IWV, which

is already accurately provided by the MWR. In addition, this result gives us confidence that the developed OEM water vapor profiles are well constrained with respect to the integrated value.

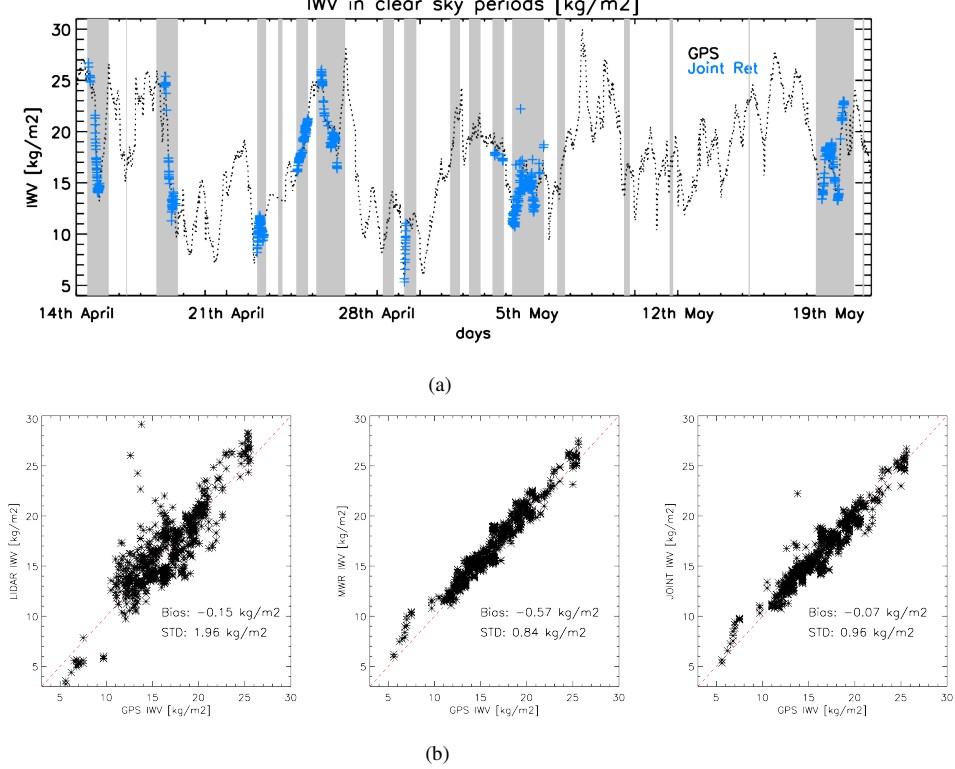

**Figure 5.** (a) Time series of IWV during the whole HOPE period for clear sky cases from: the GPS signal (black) and the one calculated from the joint retrieval, available only in clear sky cases (blue). Shaded areas represent the RL availability. (b) Scatterplot for the three cases: the joint retrieval, only MWR and only Raman Lidar (from left to right), against the GPS.




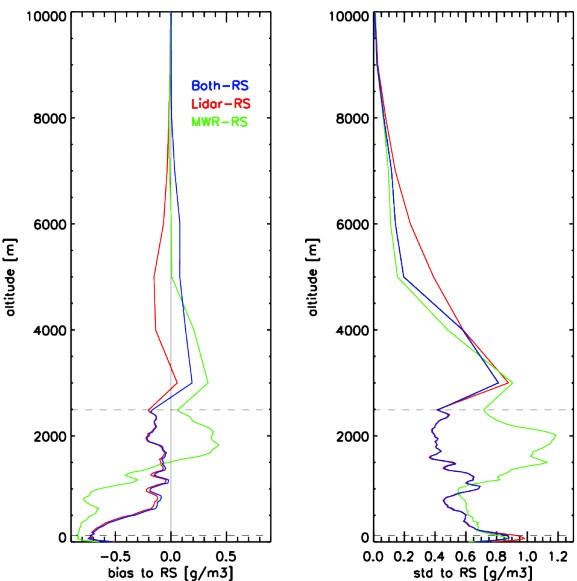

**Figure 6.** Mean and standard deviation of the difference between the 18 clear sky radiosondes: MWR (in green), RL (in red) and the combination of both (blue). The dashed horizontal lines enclose the region where the lidar data is used.

## 5.2 Comparison to RS

As explained above, the retrieval grid of each profile depends on how much data from the RL can be taken into account, which will depend on the atmospheric conditions, day/night, background noise, etc. In order to clearly assess the benefits of the sensor synergy, a different retrieval strategy is used for the subsequent tests: the algorithm is applied using only the RL profiles up to a fixed altitude in order to retrieve all profiles using the same vertical grid. Thus, all RL profiles have been capped at an altitude of 2.5 km. In the case that a given lidar profile gets too noisy before this altitude, the profile is discarded and not taken into account for the statistics. This cut-off-altitude is chosen in order to keep at least 75% of the profiles within the statistics (only 23% of the considered RL profiles reach 100% relative uncertainty at a height lower than 2.5 km). This strategy simplies the separate study of three atmospheric regions, defined as follows:

- Region a) from ground to 180 m: no lidar data is available

- Region b) from 180 m to 2.5 km: the only domain where there is lidar data. It is enclosed inside the dashed horizontal lines in Fig. 6.

- Region c) from 2.5 km to 10 km: no lidar data is considered.




At first, a comparison of the absolute humidity profiles against the radiosonde profiles is performed. Unfortunately, only 18 valid clear sky radiosondes have been found during the periods where BASIL measured. In Fig. 6, the bias (on the left) and the standard deviation (on the right) to the RS are presented for the three cases: only-MWR, only-RL and the MWR+RL combination.

Region (a), exhibits the largest standard deviations (STD) and biases, with similar values for the three cases. In addition to the fact that no lidar data is available here, this result may be due to different surface-related local effects at the site where the RS was launched ($\sim 4km$ distance) and at the site where the instruments measure. In addition an insufficient venting of the RS in the lowest 100 m may act as an additional uncertainty.

In region (b), bias and standard deviation for the only-RL and RL+MWR are very similar, whereby only-MWR reveals the largest values. The similarity between only-RL and the combination is again explained by the small uncertainty associated to the lidar measurements. The product of the combination tends to the lidar data when available, as seen in section 4.1. From $\sim 500m$ to 2.5 km, both only-RL and RL+MWR show a small bias on the order of $\sim 0.2 g/m^3$, but below this altitude, the deviation increases up to $\sim 0.75 g/m^3$. This fact may suggest that the lidar data in the lower 500 m could have some additional issues with the RL OVF. This feature will be examined in more detail in subsection 5.5.

In region (c) all the three values for the different retrievals are similar. The only-MWR seems to perform best when comparing to the RS, because both its bias and STD are the smallest. The only-RL case presents the largest bias and STD because in this region only information from the a priori is provided. The combination of the two sensors presents intermediate values, however, more similar to the only-MWR case.

Unfortunately, this set of only 18 radiosondes does not allow a significant assessment of the synergy benefits. In addition, when interpreting the results in Fig. 6, one must take into account that the RS itself presents some sources of uncertainty which are not easy to quantify, e.g. the launch distance of 4 km to the instrument site, drifts of the balloon, dry bias (Miloshevich et al., 2001), etc. Because of that, other parameters are needed to further evaluate the synergy advantages. One quantity with this capability is the theoretical OEM uncertainty of the retrieved profiles (see eq. 4). This parameter is studied in the following subsections.

### 5.3 Theoretical error comparison

As already mentioned in section 3, the algorithm provides an estimation of the a posteriori error for the retrievals, see eq. (4). For each profile the associated theoretical error profile is computed in the three different cases: using only-RL, only MWR and the RL+MWR combination.

In order to investigate the algorithm performance during day- and nighttime separately, Fig. 7(a) shows the mean theoretical errors for the three algorithm setups, differentiating between day-time and night-time. Note that, in this study, no clipping is performed in the measurements, and thus, we



cannot distinguish three regions according to lidar availability. This region separation will be used again in the next sections.

The lidar performance is much better during nighttime, when more than 50% of the lidar data reach a maximum useful altitude of around 7 km. The theoretical error during night is also lower than during daytime (i.e. about a factor of three smaller at an altitude of 4 km), as expected. During daytime, the highest useful lidar height reaches only a maximum altitude of around 5.5 km (Fig. 7(b)). In addition, only half of the profiles reach values higher than 3 km. Under these situations, the

MWR information is expected to be a more powerful supplement to the lidar information. This is well seen in the improvement of the theoretical error due to the addition of the MWR information that improves the theoretical error by approximately a 25% in the altitude range between 3 and 5 km. The only-MWR case remains almost invariable, because the instrument performs the same under different light conditions.

Following the same argumentation as in Section 5.2, we now perform a theoretical error analysis where all lidar measurements are capped at 2.5 km. The clipping at this altitude allows us to perform statistics with 75% of the profiles. Again, we separate three atmospheric regions (se section 5.2) where we can clearly distinguish whether RL data is available or not. This allows us to specify clearly the relative impact of MWR and lidar in the different OEM retrievals, as well as to perform

robust statistics.

Fig. 8 presents the a priori uncertainty, as well as an average over the 636 theoretical error profiles calculated after running the OEM for all the HOPE clear sky periods. Clearly the uncertainty associated to the a priori is the largest, as it represents the atmospheric variability within the HOPE period. When only the TBs of the MWR are introduced in the algorithm, the average error estimate is re-

duced at least by half throughout the whole atmosphere with respect to the a priori uncertainty. When only the lidar information is used by the algorithm, the error in region (b) is strongly reduced with respect to the other two previous cases. Compared to the only-MWR error, which has an average of $\sim 0.7 g/m^3$, the only-RL is lowered to almost $0.1 g/m^3$. In regions (a) and (c) the only-RL error is larger than in region (b) because no lidar data is available and thus only the a priori information is

used to complete the profile. The only-RL uncertainty is indeed especially large above 3km, where it tends to the a priori uncertainty, presenting even larger values than the only-MWR error.

However, when the combination of RL+MWR is performed, the resulting error is the smallest for all the altitudes. In region (b), the error is almost the same than for the only-RL case. Outside this region, the MWR contribution plays an important role to reduce the uncertainty. In region (c),

from average uncertainty values of 0.17 and 0.22 $g/m^2$ for only-MWR and only-RL respectively, the uncertainty of the combination is reduced to an average value of 0.12 $g/m^2$. Similarly, in the lowest region, the average error for the combination is 0.30, in comparison to 0.71 and 0.33 $g/m^2$ for the only-RL and only-MWR cases, respectively. In conclusion, there is an obvious improvement in the theoretical error due to the synergy of the two instruments.





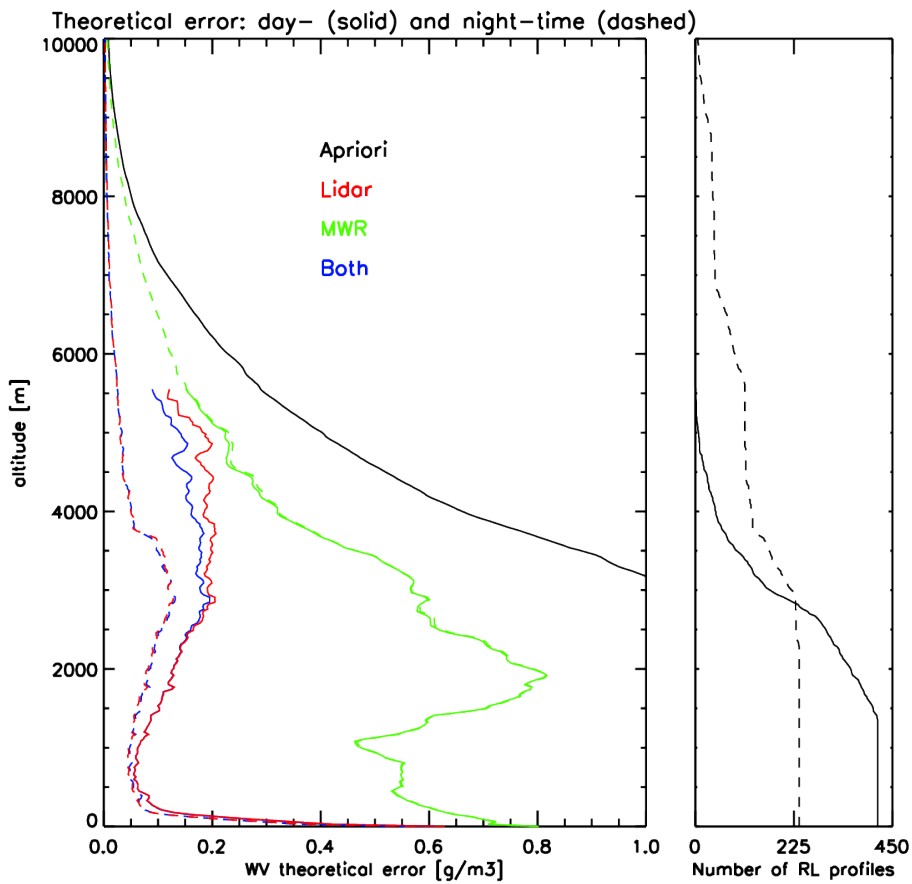

**Figure 7.** Left: mean theoretical error over the 636 clear sky cases during the complete HOPE period, separated into daytime (solid) and nighttime (dashed) measurements. In black: a priori uncertainty (lowest 3 km are out of margins). Red: only-RL. Green: only MWR. In blue: the MWR+RL. Right: number of RL profiles reaching each altitude, corresponding to the number of profiles used to calculate the average in the left panel.





One can quantify the relative error reduction $err_{red}$ of the joint retrieval in comparison to the instruments working alone. We can calculate this value as the difference between the single-instrument and joint theoretical error profiles, divided by the single-instrument one. That is:

$$err_{red,i} = \frac{err_i - err_{joint}}{err_i} \cdot 100 \qquad (10)$$

where $i = [RL, MW]$ and represents the averaged error profiles for the two different scenarios:
when only-RL and only-MWR is used (Fig. 8). Then, $err_{red,i}$ is a profile representing a relative error reduction as a function of the altitude. The average error reduction for the absolute humidity in the complete atmospheric profile is 60% (38%), with respect to the retrieval using only-MWR data (only-RL). This improvement is especially clear in region (c), above the available lidar data. The improvement of the combination in region (a) is better analysed with the experiment in the
subsection 5.5.

## 5.4  Degrees of freedom

Another parameter to assess the retrieval performance is the DOF (see Sec. 3.1). DOF allow us to study the amount of information provided by the different instruments in the three different atmospheric regions described in section 5.2. Fig. 9 represents the vertical profile of cumulative degrees
of freedom (CDOF) for the different instrument combinations, obtained as an average over 636 profiles. In the case of only-MWR, the CDOF are smaller than for the other cases, reaching a maximum of 2.26 at 10 km, in agreement with previous studies (Löhnert et al., 2007). Whenever lidar is available, the CDOF increase linearly, due to the independent information of each altitude bin measured by the lidar. In the case of only-RL, above 2.5 km the cumulative DOF remain constant because no
additional information is introduced. However, for the RL+MWR the CDOF increase above 2.5 km thanks to the inclusion of the MWR measurements. Tab. 1 summarizes the values in Fig. 9. For the only-RL case: in the regions where no lidar data is available ((a) and (c)), the DOF are, as expected, zero. In region (b), the total number of average DOF are around 26 which means that the lidar with the assumed $S_e$ and the constraint provided by $S_a$ provides 26 independent pieces of information
for humidity profile retrieval. The total average number of DOF in the column is largest for the combination of the two instruments, increasing in almost 2 DOF with respect to the only-RL case. The numbers for the MWR+RL combination show that the inclusion of MWR results mainly in an increase of DOF (+1.6) in region (c), whereas in region (b) the DOF remain almost the same. This implies that large parts of the DOF contained in the only-MWR retrieval for the complete profile
(2.26) have now been shifted to the region above 2.5 km. This optimal exploitation of the MWR information content due to constraints set by the lidar in other altitude regions clearly shows the synergy benefit.





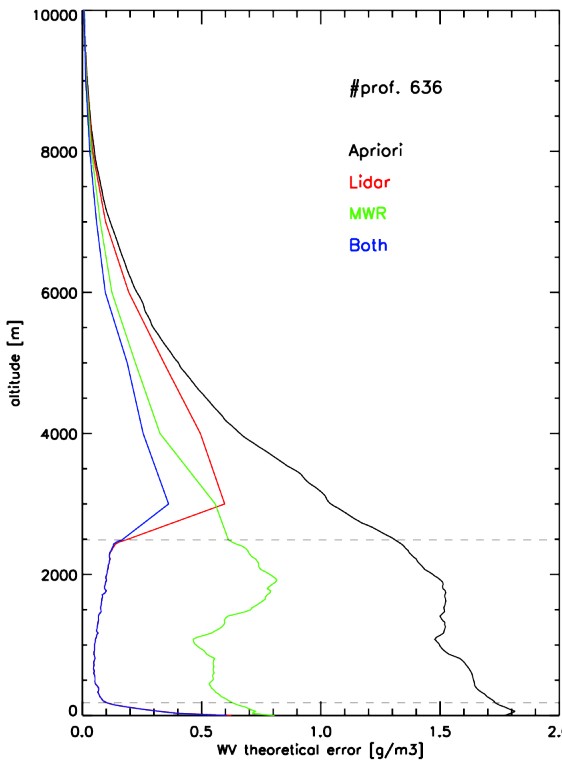

**Figure 8.** Mean theoretical error over 636 clear sky cases during the complete HOPE period. The lidar data has been artificially cut-off at 2.5 km. In black: a priori uncertainty. Red: only-RL. Green: only MWR. In blue: the MWR+RL. The dashed horizontal lines enclose the region where the lidar data is used.

### 5.5 Sensitivity study in the lower atmosphere

As argued in section 5.2, the high bias values for only-RL and RL+MWR from ground to 500 m (Fig.
5 6) might reveal a problem with the lidar OVF in this region. To asses the retrieval performance in
the case of a larger non-overlap region, we run the retrieval considering that the OVF of the RL does
not allow us to obtain valid measurements from the lowest 500 m, instead of 180 m. Thus, lidar data
from 180 to 500 meters is discarded in all the profiles. The algorithm is run again for the complete
HOPE period taking this condition into account.
Fig. 10 shows the mean theoretical error for the expanded zero-overlap region (ZOR) together
with the initial ZOR (up to 180 m). In both cases (regular ZOR and increased ZOR), the results are
very similar in regions where the RL data is available (from 500 m to 2.5 km), with the theoretical
error of the MWR+RL matching that of the only-RL. However, in the lower region of the increased





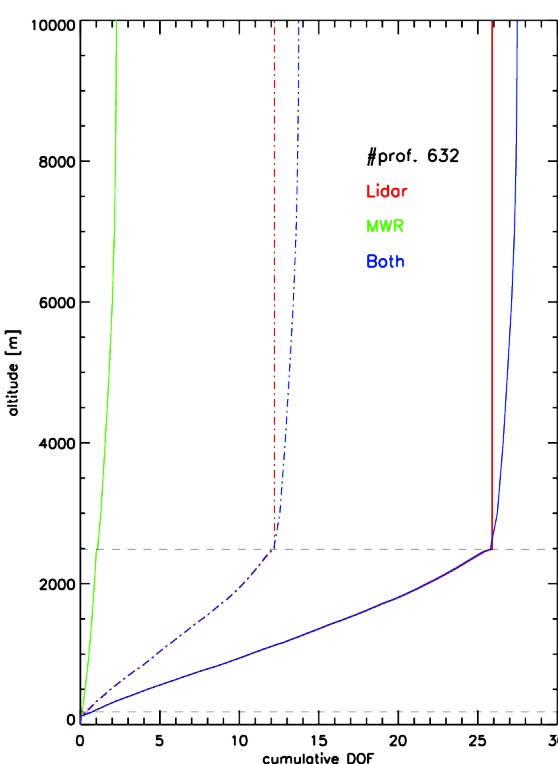

**Figure 9.** Cumulative degrees of freedom per profile for the different instrument combinations: in red, only-RL; in green, only-MWR and in blue, MWR+RL. The dotted-dashed lines represent the degrees of freedom for the case where the RL uncertainty has been multiplied by 4. The average number of DOF in every region are summarized on Table 1. The dashed horizontal grey lines enclose the part of the atmosphere where lidar data has been considered. The number of elements in the measurement and state vectors are 77 (66 for the dashed case) and 91, respectively.





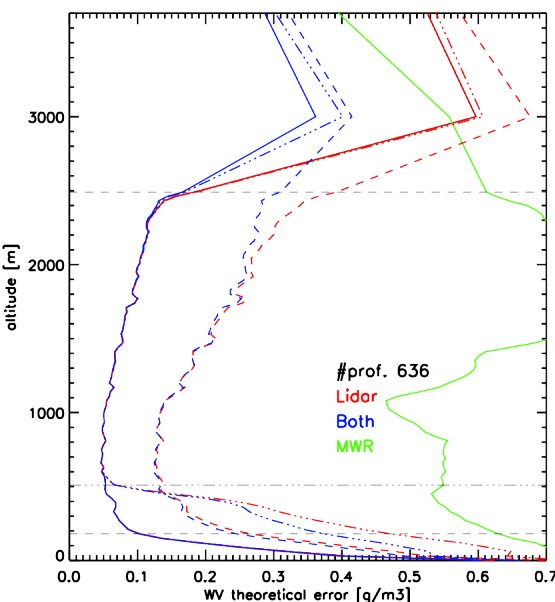

**Figure 10.** Mean theoretical error over 636 clear sky cases during the complete HOPE period. Red: only RL has been introduced in the algorithm. Green: only-MWR. In blue, the combination of RL and MWR. The dashed horizontal black lines define the region where lidar data has been considered available. The dashed red and blue lines represent the result when the lidar uncertainty has been incremented by a factor of four. The dotted-dashed red and blue lines correspond to the case where lidar data has been suppressed from ground until 500 meters. Solid lines show the errors without increments, as shown in Fig. 8.

ZOR, the MWR+RL error is smaller than the only-RL case: there is an uncertainty reduction at the

ground level of about $0.1 g/m^3$, which is gradually reduced towards the region where RL data are available. This result nicely shows the synergy benefit of both instruments in the atmosphere below 500 m. Above this point and up to 2.5 km, the error is almost equal for the cases of initial ZOR and increased ZOR. From 2.5 km to 10 km, the increased ZOR shows a slight increase in theoretical error of $\sim 0.05 g/m^3$ and $\sim 0.02 g/m^3$ for the RL+MWR and only-RL cases, with respect to the

initial ZOR. This is because the MWR information content is redistributed and more efficiently used in the lower layers of the atmosphere.

**5.6    Increase of the RL error**

In section 3.3 the components of the covariance matrix $S_e$ have been determined to our best knowl-
edge. However, it might be possible that additional uncertainty sources exist. In order to better un-



| Region | RL | MWR | Combination |
|---|---|---|---|
| a) Ground to 180 m | 0.00 | 0.07 | 0.03 |
| b) 180 m to 2.5 km | 25.90 | 1.01 | 25.75 |
| c) 2.5 km to 10 km | 0.00 | 1.18 | 1.69 |
| Total | 25.90 | 2.26 | 27.47 |
| Region | RL | MWR | Combination |
| a) Ground to 180 m | 0.00 | 0.07 | 0.06 |
| b) 180 m to 2.5 km | 12.19 | 1.01 | 12.11 |
| c) 2.5 km to 10 km | 0.00 | 1.18 | 1.57 |
| Total | 12.19 | 2.26 | 13.74 |

**Table 1.** Degrees of freedom for signal comparison for absolute humidity. Average over 636 profiles. The atmosphere is separated in three regions according to lidar availability. The DOF are presented for three cases: only RL, only MWR and the combination of both instruments. In the upper part, no increment on the RL uncertainty has been considered. In the bottom part, the RL uncertainty has been multiplied by a factor of four.

derstand the impact of the lidar uncertainties, we performed a sensitivity study increasing the lidar uncertainty.

The increase in RL measurement uncertainty is chosen kind of arbitrary based on the discrepancy between the theoretical error ($0.1g/m^3$, Fig. 8) and the mean deviation to the RS ($0.4g/m^3$, Fig. 6) at around 2 km, showing that the deviation to the RS is four times larger than the originally assumed

error. Therefor, we have increased the RL uncertainty by a factor of 4 to study the sensitivity of the retrieved profile error with respect to the RL measurement uncertainty.

The results of this test are plotted in Fig. 10, together with the initial values (without increment), for the only-RL and MWR+RL cases. The new averaged errors are a very similar at the ground, but they have increased by a factor of 2 to 3 in region (b). The uncertainty is less than a factor

of 4 because of the stabilization by the prior. In case of increased RL uncertainties, the difference between the errors of the only-RL and RL+MWR (dashed lines) is more noticeable than in the original case (solid line), especially from 2 km upwards. Note that already at 2.5 km, the error reduction for including the MWR, reaches values close to $0.1g/m^3$. Thus, as expected the synergy benefit increases.

Also, when an increment in the RL uncertainty is considered, the amount of useful information provided by this instrument is smaller, and thus the DOF are reduced. This reduction can been seen in all regions where the RL is involved (Fig. 9). In case of an uncertainty increase of a factor of 4 the total average DOF are reduced by a factor of $\sim 2$ (Tab. 1). Note that, naturally, the DOF values for the MWR only retrieval remain the same.



The results presented so far confirm that the RL+MWR water vapor synergy is meaningful and advantageous. In addition, they suggest that a careful specification of the instrument uncertainties, specially for the RL, is required.

## 6 Conclusions

Atmospheric humidity is an essential variable for the description of any meteorological process.
Highly resolved, accurate and continuous measurements of this parameter are required for a deeper understanding of many atmospheric phenomena. However, nowadays there is no single instrument, which can provide all of the following requirements simultaneously: complete vertical coverage, high vertical and temporal resolution of the atmospheric humidity profiles and satisfactory performance under all weather conditions. This is why the synergy of different sensors has become come
more and more into focus in the last years.

In this paper, we present a new and robust method to combine water vapor mixing ratio Raman lidar profiles and multifrequency brightness temperatures from a microwave radiometer. The joint algorithm that combines the two sensors is based on an Optimal Estimation Method, and can be also applied to measurements from one instrument alone. Results for 53 hours of clear sky measurements
during the HOPE period are presented for absolute humidity profile retrievals.

The improvements of the synergy have been analysed in terms of several parameters, like the reduction of the theoretical error or the increase of DOF, showing significant advantages with respect to the two instruments working separately. For example, when applying the combined retrieval to the complete HOPE period, the absolute humidity error can be reduced by 60% and 38% on average,
with respect to the retrieval using only MWR data or only RL, respectively. The synergy presents its strongest advantages in the regions where RL data is not available, whereas in the regions where both instruments are available, RL dominates the retrieval.

With the expansion of the ground based network of atmospheric profiling stations the application of the OEM at several sites under different climate conditions will become possible. In this respect,
the definition of an appropriate background uncertainty covariance needs to be carefully addressed. Further studies will extend the algorithm to cloudy cases and to temperature and relative humidity profiling. In addition, the method will be applied, not only to ground based measurements, but also to airborne data (Mech et al., 2014), which will allow to complete the study of meteorological phenomena from the airborne point of view.

*Acknowledgements.* Acknowledgements: This research has been financed by ITARS (www.itars.net), European Union Seventh Framework Programme FP7: People, ITN Marie Sklodowska Curie Actions Programme under grant agreement no 289923. The authors would like to acknowledge the Federal Ministry of Education and Research in Germany (BMBF), who, through the research programme *High Definition Clouds and Precipitation for Climate Prediction $HD(CP)^2$*, financed HOPE. Special thanks to Kerstin Ebell (for her important contri-





bution to the early stages of the project), Dave Turner (for his always fruitful ideas) and Bjorn Stevens (for his
useful advice).



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
