# Peer review of "Ground Based Lidar and Microwave Radiometry Synergy for High Vertical Resolution Absolute Humidity Profiling."

_Atmospheric Measurement Techniques, 2016_

## Referee Comment (RC1) · Anonymous Referee #1 · 23 Mar 2016

Retrievals of absolute humidity from a microwave radiometer (MWR) using an optimal estimation (OE) technique are enhanced by including a Raman lidar profile in the measurement vector. The vertical resolution and accuracy of the MWR profile are improved. A priori information is drawn from local radiosonde observations. The performance of the algorithm is demonstrated with data from the HOPE field campaign, exploring the influence of the two data sources both separately and together. The combination is shown to reduce the discrepancy compared to coincident radiosonde launches and GPS observations of integrated water vapour.

Having reviewed the previous submission of this paper (http://www.atmos-meas-tech-discuss.net/amt-2015-63/), I am surprised by how few changes

have been made. Parts of the text have been streamlined to present a more coherent argument, the input data is now pruned by considering the measurement uncertainty, and a validation of integrated water path includes some fairly convincing scatterplots. However, the text and figures are often identical (leaving about half of my previous comments unaddressed).

I still cannot recommend the publication of this paper, but I am not opposed to it's publication if other reviewers favour it. In essence, I find this an unsatisfyingly simple technique with far too little information available to justify the resolution and algorithm used. However, comments on the previous paper convinced me that vertically resolved humidity measurements are sufficiently poorly constrained that any validated attempt to improve their resolution is a step in the right direction. This algorithm meets that standard and the improved text makes it clearer that this is primarily a microwave retrieval that takes advantage of available lidar data.

The understanding of the subtleties of optimal estimation theory is poor but, as it is a complex theory that is widely misused, I cannot hold that against the authors. If the paper is published, the following issues must be corrected:

- The discussion of the averaging kernel in lines 219 to 228 is wrong. Quoting p. 47 of Rogers (2000), "rows of $\mathbf{A}$ are generally peaked functions . . . with a half-width which is a measure of the spatial resolution of the observing system, thus providing a simple characterisation of the relationship between the retrieval and the true state."

  On line 220, the averaging kernel actually describes the final values' dependency on their true magnitude, in this case indicating the smearing of information across multiple levels. The subspace of state space in which the retrieval must lie is constrained by the a priori covariance matrix.

  I have never encountered the use of degrees of freedom to measure vertical resolution, as in Eq. (6). After conferring with colleagues that work more closely

with OE of thermal profiles, we consider it to be at a confusing and poor choice of metric, if not intrinsically wrong. A much more common and robust metric would be the width of the averaging kernels (e.g. the full-width at half maximum of a row of $\mathbf{A}$).

- You still misuse the term 'error' in Section 5.3. In layman's terms, the error is how wrong your measurement was and the uncertainty is how wrong you think it might be. Robust definitions can be found at http://www.iso.org/sites/JCGM/ GUM-introduction.htm. The error can be approximated by considering the difference between the retrieved value and a more accurate reference measurement, as you do in Fig. 5. What OEM estimates is the uncertainty on the retrieved value, which describes the range of errors you would expect to see if you infinitely repeated the observation.

  I also understand why you add the word 'theoretical', but it isn't necessary as the uncertainty is a prediction of a probability distribution.

A few more minor points and comments:

- TB is a non-standard (and frankly annoying) abbreviation for brightness temperature. I would suggest BT or $T_B$.

L145 The word 'drift' doesn't appear in that Whiteman paper. I think you mean the thermal sensitivity of the filters.

L150 There is a subtle point here that, though you don't need to mention it in the paper, you may wish to consider. Poisson statistics state that the variance of a measurement sample is equal to its mean. The lidar community uses this to assume that the value of a measurement is equal to its uncertainty squared. However, that measurement is only one sample from the distribution and is therefore an imperfect estimate of the mean; it's simply the best estimate available. This isn't usually

important but in a statistical analysis, such as OEM, this approximation implicitly states that smaller values are more accurate as they have smaller uncertainty (e.g. the OEM will fit $1 \pm 1$ more closely than $100 \pm 10$). Hence, you may wish to investigate if your analysis is biased towards small values in the presence of exceptionally negative noise (i.e. data noticeably smaller than that around it).

L208 The test this sentence describes doesn't match the condition given in Eq. (3). Which one do you actually use?

L234 If you're using the radiosonde data both to determine the lidar calibration factor and as an a priori, why don't you put the calibration factor in the state vector and constrain it (and it's uncertainty) with the a priori? For example, the difference between the blue and black curves in Fig. 2 is about 5%, which would be accounted for by the uncertainty in the calibration factor.

L245 This reviewer is pleased to see correlation matrices rather than covariances.

§3.4 I remain disappointed that you do not consider a more detailed forward model for the lidar. Could this be mentioned as possible future work, in an attempt to inspire other researchers?

Fig. 2 The error bars don't cover the discrepancy between the retrieval and the radiosonde. Does this mean that your uncertainty estimate is too small or is the uncertainty on the radiosonde data large enough for the two profiles to be consistent?

L335 The value at 5 km is consistent with those at 3 and 4 km, so you can't necessarily call that an increase. Your argument is strongest when pointing out that the joint technique gets 3 km closest to the radiosonde.

While the English isn't native, I find it comprehensible and appreciate the thorough descriptions and arguments. I would recommend the lead author spend an afternoon

proof-reading the paper, especially the middle sections, as there are a number of mis-spellings and duplicated words. The technical corrections I caught while reading follow:

- I do not know if this journal prefers 'ground-based' to be hyphenated; you use both so please pick one.

- Units are frequently italicised (presumably because they have been included within a $$ environment). Please consistently use plain font.

- Many of the references list both a DOI and a URL; the URL is redundant.

L2 Nowadays there are a wide

L31 which is difficult to capture with one instrument

L41 Perhaps use 'have become' rather than 'became' and 'over recent years' rather than 'during the last years'.

L45 You use 'day time' here and 'daytime' on line 49. Please pick one.

L59 Are you sure you mean 'features'? I thought the measurements of a MWR would be better described as 'levels'.

L83 'to incorporate' could be removed without changing the meaning of this sentence. If you prefer to keep it, 'one' needs to precede it.

L85 You don't need to pluralise 'month' when used as an adjective.

L111 Raman scattering of the 355 nm beam

L133 During HOPE, BASIL was calibrated

L134 calibration coefficient was estimated by comparing

L144 Use `H$_2$O` rather than `$H_2O$`. Repeated on L165 and L167.

L163 of the K-band contain

L166 liquid water increases with

L195 This equation doesn't conform to the journal's style guide.

L197 the MWR and the profile of the mixing ratio

L203 when a perturbation is added to the atmospheric state vector

L222 degree of freedom and can be interpreted as

L215 represent the number of independent

L295 divides the atmosphere into layers

L298 To typeset the second exponential, I would recommend
`\exp\left( -\int_0^s \alpha(s') \mathrm{d}s' \right)`

L317 a complete profile from the ground up

L324 a dominant role in defining the vertical

L329 The uncertainty is small in the region

L350 the vertical resolution for only-RL becomes infinite.

L381 during HOPE, and therefore this period

Fig. 5 Invert the order in which the three plots of 5(b) are described to mimic the left-to-right manner in which they are presented.

L462 regions (see section 5.2)

L510 The average total number of DOF

L511 increasing by almost 2 DOF

L542 The magnitude of the increase in RL measurement uncertainty is based on the

L545 error. Therefore, we have

L548 The new averaged errors are very similar

L569 different sensors has come more and more into focus

L616 The page number for Delanoe and Hogan (2008) is D07204.

L619 The page number of Di Girolamo et al. (2004) is L01106.

L656 The page number of Löhnert et al. (2007) is D04205.

L662 The page numbers of Löhnert et al. (2014) are 1157–1174.

L680 An extraneous BibTeX field appears to have been printed between the DOI and year.

---

## Referee Comment (RC2) · Anonymous Referee #2 · 24 Mar 2016

The manuscript amt-2016-46 by Barrera-Verdejo et al. presents a method to combine water vapor measurements from two ground-based instruments, a Raman lidar (RL) and a microwave radiometer (MWR).

Sensor synergy is particularly important as it helps overcoming the limitations of currently available remote sensing instruments (such as RL and MWR) towards the fulfillment of the user requirements set by WMO for meteorology and climate applications. Atmospheric humidity is one such an essential variable that currently miss to meet the WMO requirements, specially for vertical resolution.

The proposed method shows the potential of RL and MWR synergy for increasing humidity profiling, both in terms of accuracy and vertical resolution, with respect to the

observations from the two separate instruments. In addition, the manuscript evaluates the effects caused by incorrect estimate of the instrument specification, such as the RL overlap function and measurement uncertainties.

I believe the manuscript is worth publication. I only have the few minor comments below, which I believe would improve the manuscript, if properly addressed.

- L291: The authors choose to use as input to the Optimal Estimation Method (OEM) a mix of retrievals (RL absolute humidity) and direct observations (MWR Tb). I'm not against this approach, but I suggest the authors explain the reason behind this choice in Section 3.3

- L313: Maybe this is stated before, but it would be worth reminding here: how was the prior knowledge estimated?

- L396: there are few cases in which the joint IWV differs more than MWR only from GPS. Particularly one point at 22 kg/m2 while GPS is 14 kg/m2, where it seems that the joint retrieval has followed the lidar IWV. Could you comment on the nature of these few cases?

- Figure 2: The caption miss to explain what the horizontal blue lines mean. I guess these indicate the estimated retrieval error, but it should be explained.

- Figure 4: It would be useful to see the same time-height cross section as seen from the two separated instruments (preferably with the native retrievals) to appreciate visually the added value of synergy.

- Figure 5: Panel a: It says clear-sky data only are presented, but I don't see any gaps for cloudy-sky periods. Gaps were shrunk or is it a 25-day clear-sky period? Not clear. Please explain in the caption. Panel b: I guess the first and last panels should be switched. Or alternatively the caption should say from right to left.

Typos:

- L50: calibrated.This

- L62: where there lidar data

- L89-91: section -> Section (or the other way around, just pick one)

- L258: way to due lack

- L263: can to lead

- L285: "in different heights" -> "at different heights"

- L330: be expected 2.

- L381: therefor

- L406: simplies -> simplifies?

- L478: than -> as

- L526: regular -> initial

- L545: Therefor
* * *

---

## Referee Comment (RC3) · Anonymous Referee #3 · 28 Mar 2016

Review of paper: amt-2016-46 Ground based lidar and microwave radiometry synergy for high vertical resolution absolute humidity profiling.

The paper combines LIDAR and MWR measurements to improve vertical profiles of humidity between the ground and the upper troposphere. It applies an optimal estimation technique to evaluate the single-instrument and combined retrievals.

I have a few reservations about the paper and the results. The first reservation is of a general nature. The RL is obviously a superior methodology to the MWR for humidity retrievals. Although the authors use the MWR measurements to improve the RL retrievals where retrievals from the latter are not reliable, the MWR retrievals themselves have very little information above the first kilometer.

[Figure]

It is possible that the RL profiles could be improved above the boundary layer just by choosing a better climatology or may be a model output without going to the extent of doing an optimal retrieval estimation. The only place where I could see some real advantage of using MWR measurements is in the lowest 200-500 m although in the case shown the results appears mixed.

The second reservation is about the conclusions as I am not sure that the results entirely support the conclusions.

Specific comments:

Page 8: Line 225, Eq. 6: Can the author provide a reference for this definition of vertical resolution? Usually the Backus-Gilbert technique is used to define the vertical resolution from the spread function. An example of application of this technique to determine the microwave radiometer vertical resolution can be found in Westwater and Snider and Carlson (J. Appl. Meteor., vol. 14, pp. 524–539, 1975).

Page 16 Fig. 6. The results in Figure 6 are mixed. In the upper troposphere the RL seems to have the lowest bias up to 4 km. Above 4 km the combined retrievals show a very small improvement, however what the standard deviation is considered I am not sure that the improvement is clear.

Page 17, section 5.3 and 5.4 I am not sure what is intended by theoretical error shown in Fig. 7. I think the author means the "a posteriori" covariance. However this measure of uncertainty, although necessary, represents a partial picture. A better estimate of "error" intended as RMS Error is the one you provide in the comparison with radiosondes in Fig. 6. The authors should probably change "theoretical error" with "covariance" if this is what they meant. Otherwise they should explain what they mean by "theoretical error". It is not clear how the uncertainty shown in Fig. 7 and 8 relates to the error bars shown in Fig. 2 (the text says they are both computed from Eq. 4), however the values seem considerably different. In particular the error bars above 2 km in Fig. 4 seem to be ∼0.5 g/mˆ3, but they seem smaller in Fig. 7. Or it is just due to the different

scales of the plots? I am not sure I entirely understand the difference in what is plotted in Fig. 7 and 8, besides the classification between daytime and nighttime. Could you please explain that more clearly?

Page 25 line 576: "The improvement of the synergy have been analyzed in terms of several parameters like the reduction of the theoretical error or the increase in DOF, showing significant advantages..." I am not entirely sure about the accuracy of this statement. The theoretical error (which is the a posteriori covariance) is related to the DOF. The two metrics are not independent and essentially convey the same information in different form. Although it is true that the analysis shows the reduction of the covariance after the retrieval, the comparison with the radiosondes conveys mixed messages about the actual usefulness of the MWR measurements.

Overall the paper provides useful information but the discussion can be improved, therefore I suggest major revisions.

There are a few English corrections needed: Page 24 line 542 "is chosen kind of arbitrary" can be rephrased: "The increase in RL measurement uncertainty is arbitrarily chosen based on..." Page 25 line 569: "...synergy of different sensors has become come more..." remove come Page 576: "several parameters like" "like" can be replaced with a colon.

---

## Referee Comment (RC4) · AH Haefele (Referee) · 7 Apr 2016

This paper is a resubmission of the discussion paper entitled "Ground based lidar and microwave radiometry synergy for high vertically resolved thermodynamic profiling" by the same authors which has been rejected for publication (http://www.atmos-meas-tech-discuss.net/amt-2015-63/). I have acted as reviewer for this former publication (Anonymous Referee #2). My review of the original manuscripts can be found online and the review of the revised manuscript is provided as supplement to this review, since it was not published online. I do of course not expect any response to this former review!

The text is now very clear and reads well. The issue of the Raman lidar (RL) covari-
ance matrix is correctly presented. I regret a lot that the averaging kernels have been removed again, it would be very interesting for the community to see averaging kernels of a combined retrieval! I encourage the authors to include averaging kernels in the response to this review and to discuss the difficulties in their interpretation. I recommend the manuscript for publication in AMT after minor revisions.

Minor remarks

L 42: Include also R. J. Sica and A. Haefele, "Retrieval of water vapor mixing ratio from a multiple channel Raman-scatter lidar using an optimal estimation method," Appl. Opt. 55, 763-777 (2016)

L 49: This is demanding but demonstrators exist. Include:

Dinoev, T., Simeonov, V., Arshinov, Y., Bobrovnikov, S., Ristori, P., Calpini, B., Parlange, M., and van den Bergh, H.: Raman Lidar for Meteorological Observations, RALMO – Part 1: Instrument description, Atmos. Meas. Tech., 6, 1329-1346, doi:10.5194/amt-6-1329-2013, 2013.

Brocard, E., Philipona, R., Haefele, A., Romanens, G., Mueller, A., Ruffieux, D., Simeonov, V., and Calpini, B.: Raman Lidar for Meteorological Observations, RALMO – Part 2: Validation of water vapor measurements, Atmos. Meas. Tech., 6, 1347-1358, doi:10.5194/amt-6-1347-2013, 2013.

L 141: Say explicitly how much the standard deviation is.

L 330: Something is wrong with "as to be expected 2".

L 350: The vertical resolution tends to infinity because the diagonal elements of the averaging kernels tend to zero. Include this explanation.

L 354: Low resolution is bad, high resolution is good!

L 391: It seems the panels of Fig. 5b are not in the right order. Reading the caption I understand 1.96 for combined, 0.84 for MWR and 0.96 for RL. The authors should also

comment on the biases.

L 445: There is no a and b in Fig. 7.

L 548: This does not sound right. It seems you scaled the variance by a factor of 4 and hence the standard deviation scales by a factor of 2. I expect in the RL region the a posteriori uncertainty if fully determined by the RL uncertainty.

Fig. 4: Mark the upper boundary of the RL data.

Fig. 7: Why do the solid lines stop at 5.5 km?

Please also note the supplement to this comment:
http://www.atmos-meas-tech-discuss.net/amt-2016-46/amt-2016-46-RC4-supplement.pdf
* * *
[Figure]

**Supplement:**

**Review of revised version of „Ground based lidar and microwave radiometry synergy for high vertically resolved thermodynamic profiling" by Barrera-Verdejo et al.**

The major changes in the revised manuscript are the addition of an Appendix showing averaging kernels and Jacobians, and the removal of the part on temperature and simultaneous retrieval of temperature and humidity. Further, the authors addressed all my comments in detail.

I acknowledge that the manuscript has improved and the response has clarified many things, but the study the way it is presented does still not meet my standards for scientific publication for the reasons presented below.

Therefore, I cannot recommend the manuscript for publication in AMT. However, I consider the work innovative and significant for the scientific community and I strongly encourage the authors to solve the remaining issues and resubmit the work for publication.

**Major comments**

Definition of measurement covariance matrix, Se, of Raman lidar (RL)

The adaptation of the text to address this issue is minimal. The authors state that they use a simplified version of Se and refer to other publications where this simplification has been used. With this, the situation is correctly presented in my view. A statement on the fact that systematic errors are present, important and correlated would have been adequate.

In the response the authors introduce various contradictions by claiming that systematic errors are not important (response p13, manuscript l134). Because at the same time the error in the calibration constant is reported to be around 5% and hence comparable to the random error, and second Section 4.2.4 is dealing with FOV effects, which shows that systematic errors are present and important. Despite this, the authors go as far as claiming in the response, that "no systematic error associated with the FOV, […] is present in the reported Raman lidar measurements".

I do also not agree with the discussion of the error correlation in the response. The statement "…one can say that there is no correlation between lidar-derived atmospheric products at different altitudes as long as no vertical smoothing is applied to the data." is not correct. I make an example: if all altitude levels have been calibrated with one and the same calibration factor, then all levels have the same uncertainty due to calibration, and hence are correlated.

Presentation of averaging kernels (AVK) in Appendix

I am very happy to see AVKs in the Appendix! However, the Appendix in its current form raises more questions than it supports the results in the manuscript. The following figures need more explanation:

Fig. 10:

- Why do the AVKs at lidar levels not reach a value close to 1? We would expect that at the levels of the lidar (0.5 km < z < 2.5 km) a change in the atmosphere would seen in the same way by the retrieval, i.e. the AVK would be close to 1.
- How come that the AVKs above 2.5 km have the same or even higher peaks than below 2.5 km? One gets the impression that the retrieval is almost more sensitive above 2.5 km.

- The AVKs in panel b) look very strange, individual AVKs cannot be distinguished and the red AVKs of higher levels look very unreasonable (same for panel c)). Here I would expect "normal" MWR AVKs for a humidity retrieval (i.e. something similar as shown here http://www.atmos-meas-tech.net/4/1891/2011/amt-4-1891-2011.pdf)

Fig. 12:

- J is the derivative of the forward model with respect to the atmosphere. Hence, why is J=0 for z<2.5 km? If one perturbs at 2 km, this has an effect on all MWR channels, no?

**Minor comments**

A3 is quite incomplete and imprecise. For example:

- Definition of vertical resolution is missing.
- What is "small resolution"? if resolution is high, the system's capability to resolve small scale features is good. Low resolution is the opposite. Here, I have the impression small resolution means high resolution.
- L568 isn't it the other way around, high layers induce small variations in the opaque channels (we perturb the atmosphere and see the effect on the observations)

Clipping of RL profile

I understand the argumentation of the authors to clip the RL profiles at 2.5 km and agree with it. However, the new Fig. 6 is not well described:

- Why do the solid lines (daytime) stop at 5.5 km? the retrieval is defined for the entire range, at least for MWR and BOTH.

---

## Author Comment (AC1) · 17 Jun 2016

**REPLY TO REVIEWER #1**

The authors highly appreciate the constructive comments. They are very useful contributions that will certainly help to improve the revised manuscript. In the following, the authors reply point by point to all Reviewer comments, which are written in italic while our replies are in standard font. Within the manuscript all changes from the submitted version are highlighted in red.

**MAJOR COMMENTS OF REVIEWER #1:**

**General remark:**
Different communities use different notations, which is also true for the lidar and microwave community. This paper should serve both communities and therefore, in order to keep things straight forward, we decided to stick as close as possible to the notation of Rodgers (2000) – the most important textbook for optimal estimation.

*1. The discussion of the averaging kernel in lines 219 to 228 is wrong. Quoting p. 47 of Rogers (2000), "rows of A are generally peaked functions . . . with a half- width which is a measure of the spatial resolution of the observing system, thus providing a simple characterisation of the relationship between the retrieval and the true state."*
*On line 220, the averaging kernel actually describes the final values' dependency on their true magnitude, in this case indicating the smearing of information across multiple levels. The subspace of state space in which the retrieval must lie is constrained by the a priori covariance matrix.*
*I have never encountered the use of degrees of freedom to measure vertical resolution, as in Eq. (6). After conferring with colleagues that work more closely with OE of thermal profiles, we consider it to be at a confusing and poor choice of metric, if not intrinsically wrong. A much more common and robust metric would be the width of the averaging kernels (e.g. the full-width at half maximum of a row of A).*

The authors consider correct what is stated in the manuscript in lines 219 to 228. To corroborate this we cite Rodgers (2000), pages 52, 53 and 54:

"*Resolution*, like *information*, is a word with a multiplicity of meanings, and tends to be used differently in different contexts. […] Possibilities include characteristics of the averaging kernel or state resolution matrix, such as the width of the averaging kernel or the point spread function, where *width* has many possible interpretations, the response of the retrieval to sine wave perturbations in the state, and **the range of heights covered divided by number of independent quantities measured**. […]
Possible characterisations of resolution are […]
(iv) the degrees of freedom for signal, $d_s$ is the trace of the averaging kernel matrix. **Consequently the diagonal of A may be thought of as a measure of the number of levels per degree of freedom, and thus a measure of resolution.** "

Which is the definition of equation (6). Moreover, other authors have previously used this definition for vertical resolution, e.g. Liu (2014).

*2. You still misuse the term 'error' in Section 5.3. In layman's terms, the error is how wrong your measurement was and the uncertainty is how wrong you think it might be. Robust definitions can be found at http://www.iso.org/sites/JCGM/ GUM-introduction.htm. The error can be approximated by considering the difference between the retrieved value and a more accurate reference measurement, as you do in Fig. 5. What OEM estimates is the uncertainty on the retrieved*

*value, which describes the range of errors you would expect to see if you infinitely repeated the observation.*
*I also understand why you add the word 'theoretical', but it isn't necessary as the uncertainty is a prediction of a probability distribution.*

We agree with the reviewer, but as explained in the general remark, we prefer to consequently adapt the notation given by Rodgers (2000). According to this book, $\hat{S}$ (or $S_{op}$ in our manuscript) is the total retrieval **error** covariance matrix (see page 58). In addition, with the word *theoretical* we want to emphasize that it is an estimate, not a direct difference to the true state.

Further, note that the term *theoretical error* is widely used in literature, e.g. Healy (2000), Phalippou (1996), Puliafito (1995), etc.

A clarification note has been included in the manuscript (former line 214), which now reads:

"From $S_{op}$, the theoretical error (in kg/m$^3$) associated to each altitude of the retrieved profile $x_{op}$ is calculated as the square root of the main diagonal elements in $S_{op}$. The word *theoretical* emphasizes that it is an a posteriori estimate, and not a direct difference to a given reference".

**MINOR COMMENTS:**

*TB is a non-standard (and frankly annoying) abbreviation for brightness temperature. I would suggest BT or T*

We understand the argument but there are some reasons for the use of TB for *brightness temperature*. First, we do not want confusion with the physical temperature that's why the do not use T. Second, microwave radiometry developed from radioastronomy, where the notation of brightness temperatures as TB stems from.

TB is used throughout many different applications of microwave radiometry, being one of the most important textbooks Janssen, 1993. To our best knowledge the majority of publications in the ground-based microwave community uses TB while for space borne applications BT is preferred.

*L145 The word 'drift' doesn't appear in that Whiteman paper. I think you mean the thermal sensitivity of the filters.*

The reviewer is right. We indeed meant the "thermal sensitivity of the filters". Specifically, we refer here to the fact that the interference filter transmittance spectrum is slightly temperature dependent. As temperature increases, the thicknesses of all dielectric layers increase and all layer indices change. These effects combine in a way that the transmittance spectrum shifts to slightly longer wavelengths with increasing temperature, with the thermal coefficient being a function of wavelength. This is now more clearly stated in the text and the sentence now reads:

"For example, an additional uncertainty (<1%) may be considered related to the use of narrowband filters, the temperature dependence of $H_2O$ Raman scattering and the thermal sensitivity of the filters (Whiteman, 2003) "

*L150 There is a subtle point here that, though you don't need to mention it in the paper, you may wish to consider. Poisson statistics state that the variance of a measurement sample is equal to its mean. The lidar community uses this to assume that the value of a measurement is equal to its uncertainty squared. However, that measurement is only one sample from the distribution and is therefore an imperfect estimate of the mean; it's simply the best estimate available. This isn't usually important but in a statistical analysis, such as OEM, this approximation implicitly states that smaller values are more accurate as they have smaller uncertainty (e.g. the OEM will fit 1 ± 1 more closely than 100 ± 10). Hence, you may wish to investigate if your analysis is biased towards small*

*values in the presence of exceptionally negative noise (i.e. data noticeably smaller than that around it).*

Poisson statistics have been applied to the lidar signals used in this paper after a careful verification of its validity and applicability. In this respect, an analysis was carried out to determine total variance profiles from high resolution water vapour mixing ratio profile measurements (temporal resolution of 10 sec and a vertical resolution of 90 m) for lidar data from the same field campaign. The autocovariance method defined by Lenschow et al. (2000) was then applied to effectively separate atmospheric variance from the noise variance in the total measured variance. This method is based on the consideration that atmospheric fluctuations are correlated in time, while instrumental noise fluctuations are uncorrelated. Profiles of the total noise error (determined as the squared root of the noise variance) affecting water vapour mixing ratio measurements obtained through this method have been compared with estimates of the water vapour mixing ratio measurement uncertainty due to shot noise derived with Poisson statistics. The shot-noise error is typically a predominant part (around 80-85 %) of the total statistical error, while the reminder part is to be attributed to other statistical error sources (among others, the detectors' dark noise error). The comparison we performed for this data set confirms that the major contribution to the total noise error originates from photon shot noise. But error estimates derived with Poisson statistics are at any altitude proportional to the noise error determined through the autocovariance method. This result confirms the correct applicability of Poisson statistics in estimating measurement error and consequently the possibility to apply results obtained from Poisson statistics in the observation uncertainty covariance matrix.

***L208 The test this sentence describes doesn't match the condition given in Eq. (3). Which one do you actually use?***

That is right. The sentence has been corrected in the manuscript as follows:

"i.e. the difference between the forward model applied to the atmospheric state at iterations n and n+1…"

***L 234 If you're using the radiosonde data both to determine the lidar calibration factor and as an a priori, why don't you put the calibration factor in the state vector and constrain it (and it's uncertainty) with the a priori? For example, the difference between the blue and black curves in Fig. 2 is about 5%, which would be accounted for by the uncertainty in the calibration factor.***

The a priori information is calculated from the complete set of radiosondes launched during HOPE (217 launches in total, at least two per day). The mean calibration coefficient for the Raman lidar was estimated comparing Raman lidar and radiosonde data, but only considering clear sky radiosonde launches when the Raman lidar was operational (60 launches in total). So, the dataset used for the a priori information is numerically different from the dataset used for the calibration of the Raman lidar. Additionally, for the purpose of the Raman lidar calibration, lidar and radiosonde data are compared in an altitude region with an extent of 1 km above the boundary layer (to minimize air mass differences associated with the distance between the lidar station and the radiosonde launching facility, approx. 4 km), while the complete radiosonde profiles are used as a priori information. So, the dataset used for the a priori information is also different in terms of vertical extent from the dataset used for the calibration of the Raman lidar. Nevertheless, while in principle it would be possible to put the calibration factor in the state vector and constrain it with the a priori, potential systematic effects associated with this approach could occur.

***L245 This reviewer is pleased to see correlation matrices rather than covariances.***

For your information, the covariance matrix is presented below. It is naturally dominated by the highest occurrence of water vapor in the boundary layer.

[Figure]

**Figure R1.1 Covariance matrix derived from 217 radiosondes launched during HOPE. Covariance is shown for absolute humidity as a function of the altitude (from 0 to 10 km above the ground) in g/m3.**

***Sec. 3.4. I remain disappointed that you do not consider a more detailed forward model for the lidar. Could this be mentioned as possible future work, in an attempt to inspire other researchers?***

Indeed, a more complex forward model can be mentioned as a possible future evolution for the considered approach. A variety of lidar forward simulators have been developed in the lidar community. The following clarification has been included:

"Therefore, the lidar FM for water vapor simply performs the conversion from absolute humidity to mixing ratio. However, the implementation of a more complex lidar forward model, e.g. the approach implemented by Sica (2016), could be considered in future studies."

***Fig. 2 The error bars don't cover the discrepancy between the retrieval and the radiosonde. Does this mean that your uncertainty estimate is too small or is the uncertainty on the radiosonde data large enough for the two profiles to be consistent?***

There are three sources contributing to the discrepancy between retrieval and radiosonde:
- i)      the uncertainty of the a priori and of the lidar and MWR measurements (which are combined into the retrieval uncertainty),
- ii)      the uncertainty of the radiosonde measurement (5% RH as reported by the manufacturer) and
- iii)      the difference in the atmospheric volume measured by the instruments (lidar, MWR and radiosonde). The magnitude of the latter is difficult to assess, as no truth about the 3dimensional distribution of the water vapor field is available. As reported in the manuscript the radiosonde was launched 4 km apart from the lidar location and drifts significantly during its ascent. This source of uncertainty can be high especially in the boundary layer (see Steinke et al., 2015) and is not represented in the error bars associated to the retrieval uncertainty.

The underestimation of the retrieval uncertainty is therefore most likely due to contributions from ii) and iii), which are not taken into account in the error bars from figure 2.

***L335 The value at 5 km is consistent with those at 3 and 4 km, so you can't necessarily call that an increase. Your argument is strongest when pointing out that the joint technique gets 3 km closest to the radiosonde.***

Indeed the sentence: "Only the combined retrieval can detect the drop in humidity at 3 km and the increase at 5 km" was not really correct as in fact the absolute humidity content keeps almost constant for the combined retrieval. As suggested by the reviewer we have changed the sentence pointing to the result that at 3 km the combined retrieval gets values closer to the radiosonde. The sentence has been changed accordingly and now reads:

"Only the combined retrieval reveals absolute humidity values in agreement with the radiosonde at 3 km".

**TECHNICAL POINTS:**

1. *I do not know if this journal prefers 'ground-based' to be hyphenated; you use both so please pick one.*
2. *Units are frequently italicised (presumably because they have been included within a $$ environment). Please consistently use plain font.*
3. *Many of the references list both a DOI and a URL; the URL is redundant.*
4. *L2 Nowadays there are a wide*
5. *L31 which is difficult to capture with one instrument*
6. *L41 Perhaps use 'have become' rather than 'became' and 'over recent years' rather than 'during the last years'.*
7. *L45 You use 'day time' here and 'daytime' on line 49. Please pick one.*
8. *L59 Are you sure you mean 'features'? I thought the measurements of a MWR would be better described as 'levels'.*
9. *L83 'to incorporate' could be removed without changing the meaning of this sentence. If you prefer to keep it, 'one' needs to precede it.*
10. *L85 You don't need to pluralise 'month' when used as an adjective. L111 Raman scattering of the 355 nm beam*
11. *L133 During HOPE, BASIL was calibrated*
12. *L134 calibration coefficient was estimated by comparing*
13. *L144 Use H$_2$O rather than $H_2O$. Repeated on L165 and L167.*
14. *L163 of the K-band contain*
15. *L166 liquid water increases with*
16. *L195 This equation doesn't conform to the journal's style guide.*
17. *L197 the MWR and the profile of the mixing ratio*
18. *L203 when a perturbation is added to the atmospheric state vector*
19. *L222 degree of freedom and can be interpreted as*
20. *L215 represent the number of independent*
21. *L295 divides the atmosphere into layers*
22. *L298 To typeset the second exponential, I would recommend*
    *\exp\left( -\int_0^s \alpha(s') \mathrm{d}s' \right)*
23. *L317 a complete profile from the ground up*
24. *L324 a dominant role in defining the vertical*
25. *L329 The uncertainty is small in the region*
26. *L350 the vertical resolution for only-RL becomes infinite.*
27. *L381 during HOPE, and therefore this period*
28. *Fig. 5 Invert the order in which the three plots of 5(b) are described to mimic the left-to-right manner in which they are presented.*
29. *L462 regions (see section 5.2)*
30. *L510 The average total number of DOF*

31. *L511  increasing by almost 2 DOF*
32. *L542 The magnitude of the increase in RL measurement uncertainty is based on the L545 error. Therefore, we have*
33. *L548 The new averaged errors are very similar*
34. *L569 different sensors has come more and more into focus*
35. *L616 The page number for Delanoe and Hogan (2008) is D07204.*
36. *L619 The page number of Di Girolamo et al. (2004) is L01106.*
37. *L656 The page number of Löhnert et al. (2007) is D04205.*
38. *L662 The page numbers of Löhnert et al. (2014) are 1157–1174.*
39. *L680 An extraneous BibTeX field appears to have been printed between the DOI and year.*

Comments from 1 to 39 have been addressed in the manuscript.

**Extra references:**

Liu, D., Lv, C., Liu, K., Xie, Y., & Miao, J. (2014). Retrieval Analysis of Atmospheric Water Vapor for K-Band Ground-Based Hyperspectral Microwave Radiometer. *Geoscience and Remote Sensing Letters, IEEE*, *11*(10), 1835-1839.

Healy, S. B., & Eyre, J. R. (2000). Retrieving temperature, water vapour and surface pressure information from refractive-index profiles derived by radio occultation: A simulation study. *Quarterly Journal of the Royal Meteorological Society*, *126*(566), 1661-1683.

Phalippou, L. (1996). Variational retrieval of humidity profile, wind speed and cloud liquid-water path with the SSM/I: Potential for numerical weather prediction. *Quarterly Journal of the Royal Meteorological Society*, *122*(530), 327-355.

Puliafito, E., Bevilacqua, R., Olivero, J., & Degenhardt, W. (1995). Retrieval error comparison for several inversion techniques used in limb-scanning millimeter-wave spectroscopy. *Journal of Geophysical Research: Atmospheres*, *100*(D7), 14257-14267.

Janssen, M. A. Atmospheric Remote sensing by Microwave Radiometry, John Wiley and Sons, Inc., 1993.

Sica, R. J. and Haefele, A., Retrieval of water vapor mixing ratio from a multiple channel Raman-scatter lidar using an optimal estimation method, Appl. Opt. 55, 763-777 (2016)

---

## Author Comment (AC2) · 17 Jun 2016

**REPLY TO REVIEWER #2**

We thank the reviewer for his helpful comments, which help to improve the paper. The reviewer's comments are written in italic while our replies are in standard font. Within the manuscript all changes from the submitted version are highlighted in red.

**MINOR COMMENTS OF REVIEWER #2:**

*- L291: The authors choose to use as input to the Optimal Estimation Method (OEM) a mix of retrievals (RL absolute humidity) and direct observations (MWR Tb). I'm not against this approach, but I suggest the authors explain the reason behind this choice in Section 3.3*

The manuscript was modified and states in Section 3.3:
"The measurement vector y is composed of the TBs from the MWR and the water vapor mixing ratio (WVMR) profile from the RL. We choose the TBs to be part of the measurement vector instead of the MWR-derived profile of humidity in order to give the OEM the freedom to distribute the water vapor information to those heights where the lidar provides no information. In addition, for future applications, it allows us to extend our algorithm to simultaneous, physically consistent retrievals of temperature and liquid water. WVMR is used as the lidar measurement (with uncertainties given in Sec. 2.1.), which allows to avoid the use of a complex lidar forward operator."

*- L313: Maybe this is stated before, but it would be worth reminding here: how was the prior knowledge estimated?*

The a priori knowledge is calculated from a set of 217 radiosondes (see line 230). A reference to the previous information is introduced now in the manuscript:

"The a priori profile is the prior atmospheric knowledge (Sec. 3.2), and also the starting point (first guess) for the algorithm iteration."

*- L396: there are few cases in which the joint IWV differs more than MWR only from GPS. Particularly one point at 22 kg/m2 while GPS is 14 kg/m2, where it seems that the joint retrieval has followed the lidar IWV. Could you comment on the nature of these few cases?*

The aforementioned IWV outlier in figure 5 (from the manuscript) presents values of 22.2 and 13.8 $kg/m^2$ for the JOINT and the GPS respectively. In this specific case, there was a problem with the error estimation of the lidar signal. As can be seen in figure R2.1, the lidar data from 3.5 km up to 4.5 km presents a unexpected/unphysical strong increase. This fact would not represent a problem for the optimal estimation as soon as the uncertainty associated to these data would be large enough. But, in this case, the error associated to the data at these altitudes has been underestimated and consequently, the retrieved atmospheric state is incorrect.

This outlier has been removed from the statistics and an updated figure 4 (here Fig. R2.2) is presented in the manuscript.

[Figure]

**Figure R2.1 Absolute humidity (g/m3) for the outlier in figure 5 (manuscript).**

[Figure]

**Figure R2.2 New figure 5 (manuscript).**

*- Figure 2: The caption miss to explain what the horizontal blue lines mean. I guess these indicate the estimated retrieval error, but it should be explained.*

Thanks for catching this. Indeed they indicate the estimated error. The explanation is now included in the figure caption:

"Absolute humidity profiles for a priori (yellow), only-RL (red), only-MWR (green) and MWR+RL (blue). The horizontal blue lines correspond to the theoretical retrieval error for the MWR+RL case. The RS is used as reference (black). The dashed horizontal grey lines enclose the region where the lidar data is used. The inset is a zoom for the region close to the ground, between 0 and 250 m."

*- Figure 4: It would be useful to see the same time-height cross section as seen from the two separated instruments (preferably with the native retrievals) to appreciate visually the added value of synergy.*

A very good visual example is presented in figure R2.3. The example corresponds to a time series in the afternoon of the 4[th] of May 2013, from 18 to 18.7 UTC.

[Figure]

(a)

(b)

(c)

**Figure R2.3 Absolute humidity time series for (a) only-MWR (statistical retrieval), (b) only-RL and (c) MWR+RL (joint retrieval).**

*- Figure 5: Panel a: It says clear-sky data only are presented, but I don't see any gaps for cloudy-sky periods. Gaps were shrunk or is it a 25-day clear-sky period? Not clear. Please explain in the*

*caption. Panel b: I guess the first and last panels should be switched. Or alternatively the caption should say from right to left.*

Unfortunately the caption was misleading. It has been rewritten as follows:

"(a) Time series of IWV during the whole HOPE period from the continuous GPS estimation (black) and the one calculated from the joint retrieval, which is available only in clear sky cases (blue)."

**TYPOS:**
*- L50: calibrated.This*
*- L62: where there lidar data*
*- L89-91: section -> Section (or the other way around, just pick one)*
*- L258: way to due lack*
*- L263: can to lead*
*- L285: "in different heights" -> "at different heights"*
*- L330: be expected 2.*
*- L381: therefor*
*- L406: simples -> simplifies?*
*- L478: than -> as*
*- L526: regular -> initial*
*- L545: Therefor*

All the typos have been corrected in the manuscript.

---

## Author Comment (AC3) · 17 Jun 2016

**REPLY TO REVIEWER #3**

The authors highly appreciate the constructive comments. They are very useful contributions that will certainly help to improve the revised manuscript. In the following, the authors reply point by point to all Reviewer comments, which are written in italic while our replies are in standard font. Within the manuscript all changes from the submitted version are highlighted in red.

**MAJOR COMMENTS OF REVIEWER #3:**

*The RL is obviously a superior methodology to the MWR for humidity retrievals. Although the authors use the MWR measurements to improve the RL retrievals where retrievals from the latter are not reliable, the MWR retrievals themselves have very little information above the first kilometer. It is possible that the RL profiles could be improved above the boundary layer just by choosing a better climatology or may be a model output without going to the extent of doing an optimal retrieval estimation. The only place where I could see some real advantage of using MWR measurements is in the lowest 200-500 m although in the case shown the results appears mixed.*

Actually, for water vapor remote sensing using the 22.235 GHz water vapor line, the weighting functions show only a weak dependence on height (see http://cfa.aquila.infn.it/wiki.eg-climet.org/index.php5/MWR_Fundamentals). However, this weak dependence caused by pressure broadening of the line, makes it possible to obtain limited information (2 degrees of freedom for signal (DOF)) on the vertical distribution of water vapor in the first place. Note, that the weighting functions near the line center actually increase from surface upwards. Thus, MWR measurements between 20 and 30 GHz (K-band) can provide water vapor information throughout the troposphere and are not limited to the boundary layer (as in case of temperature profiling). This is quantified in terms of DOF in Fig. 9, which shows that information from the MWR (accounting to roughly 1.5) is added up in the region above 2.5 km. This information is most prominent for the daytime profiles, when the lidar presents a weaker performance (Fig. 7). Furthermore, the effect is illustrated in real measurements for an individual radiosonde ascent in Fig. 2, with the superior performance of the joint retrieval at 3000 m height.

It is also important to note that the full benefit of the method will be the application to cloudy scenes where the lidar is strongly limited.

*The second reservation is about the conclusions as I am not sure that the results entirely support the conclusions.*

For the sake of clarity, the conclusions have been modified as follows:

"The improvements of merging both instrument systems have been consistently analysed in terms of both the reduction of the theoretical error and the increase of DOF. Significant advantages of instrument synergy are clearly shown above the highest valid lidar signal. For example, when applying the combined retrieval to the complete HOPE period, the absolute humidity theoretical error above ~3 km is reduced by a factor of 2 with respect to the case where only lidar is used. The addition of the MWR information to the RL results in 1.6 additional degrees of freedom per signal, which are mainly distributed in the layers above the lidar noise threshold. The synergy presents its strongest advantages in the regions where RL data is not available, whereas in the regions where both instruments are available, RL dominates the retrieval."

**SPECIFIC COMMENTS:**

*Page 8: Line 225, Eq. 6: Can the author provide a reference for this definition of vertical resolution? Usually the Backus-Gilbert technique is used to define the vertical resolution from the spread function. An example of application of this technique to determine the microwave radiometer*

*vertical resolution can be found in Westwater and Snider and Carlson (J. Appl. Meteor., vol. 14, pp. 524–539, 1975).*

There are many different ways to define vertical resolution. In addition to the example given by the reviewer, Liljegren (2004) exploited the interlevel error covariance. Here we follow the approach based on the optimal estimation as presented by Rodgers (2000) in pages 52, 53 and 54:

"*Resolution*, like *information*, is a word with a multiplicity of meanings, and tends to be used differently in different contexts. […] Possibilities include characteristics of the averaging kernel or state resolution matrix, such as the width of the averaging kernel or the point spread function, where *width* has many possible interpretations, the response of the retrieval to sine wave perturbations in the state, and the range of heights covered divided by number of independent quantities measured. […]
Possible characterisations of resolution are […]
(iv) the degrees of freedom for signal, $d_s$ is the trace of the averaging kernel matrix. Consequently the diagonal of A may be thought of as a measure of the number of levels per degree of freedom, and thus a measure of resolution. "

This is the definition of equation (6). Other authors have previously used this definition for vertical resolution, e.g. Liu (2014).

***Page 16 Fig. 6. The results in Figure 6 are mixed. In the upper troposphere the RL seems to have the lowest bias up to 4 km. Above 4 km the combined retrievals show a very small improvement, however what the standard deviation is considered I am not sure that the improvement is clear.***

We included section 5.2 to show the retrieval performance in comparison to an independent measurement of humidity profiles. As we mention in the paper, this comparison is not conclusive because it is based on a limited number of samples and the agreement in terms of the atmospheric volume sampled by the radiosonde and the remote sensing systems is not perfect. In the manuscript (line 429-433) we only state that there is, on average, in region c, a small improvement of the combined retrieval with respect to RL retrieval both in term of bias and standard deviation. We are very careful to not state any improvement for the combined retrieval in the other regions

***Page 17, section 5.3 and 5.4 I am not sure what is intended by theoretical error shown in Fig. 7. I think the author means the "a posteriori" covariance. However this measure of uncertainty, although necessary, represents a partial picture. A better estimate of "error" intended as RMS Error is the one you provide in the comparison with radiosondes in Fig. 6. The authors should probably change "theoretical error" with "covariance" if this is what they meant. Otherwise they should explain what they mean by "theoretical error". It is not clear how the uncertainty shown in Fig. 7 and 8 relates to the error bars shown in Fig. 2 (the text says they are both computed from Eq. 4), however the values seem considerably different.***
The reviewer is right. A clarification is needed and introduced in the new manuscript as follows:

"From $S_{op}$, the theoretical error (in $kg/m^3$) associated to each altitude of the retrieved profile $x_{op}$ is calculated as the square root of the main diagonal elements in $S_{op}$. The word *theoretical* emphasizes that it is an a posteriori estimate, and not a direct difference to a given reference".
***In particular the error bars above 2 km in Fig. 4 seem to be ~0.5 g/mˆ3, but they seem smaller in Fig. 7. Or it is just due to the different scales of the plots?***

The values on Fig. 2 represent the theoretical error for one specific profile, while in figure 7 and 8 the values correspond to the **mean** theoretical error, **averaged over more than 600 profiles**. That is the reason why the values for the particular example in Fig. 2 might differ from the ones presented in Fig. 7 and 8.

***I am not sure I entirely understand the difference in what is plotted in Fig. 7 and 8, besides the classification between daytime and nighttime. Could you please explain that more clearly?***

On the one hand, Fig. 7 is presented in order to illustrate the benefit of the synergy separating day and night measurements. The reason for that is that the lidar presents a better performance during nighttime periods. During these periods, the vertical contribution of the MWR is reduced to the lidar overlap region in the lowest atmosphere. Nevertheless, during daytime periods, the MWR contribution to the retrieval in higher atmospheric layers can be stronger, due to the lack of lidar data. The authors wanted to highlight this difference.

On the other hand, Fig. 8 is presented as an average over the complete HOPE period, allowing us to provide an overview through the complete campaign. Here, an artificial clipping of the lidar data up to 2.5 km has been performed. Its aim is to describe the behavior of the instrument combination with a large number of profiles, in a situation where the regions with and without lidar data availability are clearly defined. In contrast, in Fig. 7, no artificial clipping for the lidar is performed.

Several clarification sentences have been included in different parts of section 5.3. (Please, see red sentences in new manuscript version).

*Page 25 line 576: "The improvement of the synergy have been analyzed in terms of several parameters like the reduction of the theoretical error or the increase in DOF, showing significant advantages. . ." I am not entirely sure about the accuracy of this statement. The theoretical error (which is the a posteriori covariance) is related to the DOF. The two metrics are not independent and essentially convey the same information in different form. Although it is true that the analysis shows the reduction of the covariance after the retrieval, the comparison with the radiosondes conveys mixed messages about the actual usefulness of the MWR measurements.*

We do not claim that theoretical error and DOF are independent measures. They state two measures with which we are able to characterize the benefit of instrument synergy. We show both measures, which underline the consistency of our results. The main advantages of MWR and lidar synergy are shown to occur above the height of the minimum lidar sensitivity. This fact becomes clear in Figs. 2, 3, 6 (right panel), 7, 8 and 9. Indeed, Fig 6 (left panel) does not show a clear improvement of the combined retrieval considering bias error. However, it does not allow to draw a conclusion from where the bias originates. We have modified the text and now it states:

"The improvements of merging both instrument systems have been consistently analyzed in terms of both the reduction of the theoretical error and the increase of DOF. Significant advantages of instrument synergy are clearly shown above the highest valid lidar signal."

**ENGLISH CORRECTIONS:**

*There are a few English corrections needed:*
*Page 24 line 542 "is chosen kind of arbitrary" can be rephrased: "The increase in RL measurement uncertainty is arbitrarily chosen based on. . ."*
*Page 25 line 569: ". . .synergy of different sensors has become come more. . ." remove come Page 576: "several parameters like" "like" can be replaced with a colon.*

The English corrections have been corrected in the manuscript.

---

## Author Comment (AC4) · 17 Jun 2016

**REPLY TO ALEXANDER HAEFELE**

The authors highly appreciate the constructive comments. They are very useful contributions that will certainly help to improve the revised manuscript. In the following, the authors reply point by point to all Reviewer comments, which are written in italic while our replies are in standard font. Within the manuscript all changes from the submitted version are highlighted in red.

**COMMENTS:**

*The issue of the Raman lidar (RL) covariance matrix is correctly presented. I regret a lot that the averaging kernels have been removed again, it would be very interesting for the community to see averaging kernels of a combined retrieval! I encourage the authors to include averaging kernels in the response to this review and to discuss the difficulties in their interpretation.*

The averaging kernels are plotted below and an explanation is given in the following:
Because the retrieved parameter *x* is a vertical atmospheric profile, the *Ak* columns represent the information distribution of the retrieved profile as a function of the altitude. For the sake of clarity, the averaging kernels have been plotted for a AH profile retrieved with constant retrieval grid with 90 meters separation. Figure R.4.1 shows the *Ak* corresponding also to the case study in figure 2 in the manuscript, where the lidar useful data covers the region from 180 meters to 2.5 km. In addition, figure R.4.2 is presented as tool to better understand figure R.4.1. The first considers the same retrieval profile than figure R.4.1 but takes into account a diagonal *Sa* in the retrieval calculation, i.e. canceling the vertical correlations.

If the RL water vapor mixing ratio values are vertically independent, i.e. *Sop* and *Sa* are diagonal matrices, the RL information at a given height will only affect this specific altitude. In the *Ak*, this is translated into delta functions at each height were RL is available (see figure R.4.2). Instead, and because of the vertical correlation introduced by the off-diagonal elements in the a priori covariance matrix *Sa* (figure 1 in manuscript), the *Ak* columns present a smooth shape. This implies that the information from a given atmospheric layer is redistributed in altitude, affecting to the neighboring regions (figure R.4.1).

The only-MWR provides much lower information content, i.e. one order of magnitude smaller than the RL, as expected. If the a priori covariance matrix is diagonal, the strongest information content will be expected close to the surface, where the instrument is more sensitive (see figure R.4.2). Nevertheless, due to the altitude correlation defined by *Sa*, the information content is re-organized in the atmosphere, showing its maximum at ~2 km, i.e. the typical boundary layer height. The *Ak* for the MWR+RL combination shows in both cases, i.e. figure R.4.2 and R.4.1, how the information content of the two sensors is optimally combined.

[Figure]

**Figure R.4.1: From left to right: averaging kernels of only-RL, only-MWR and MWR+RL. Each color corresponds to a different altitude: ground is represented by black, higher altitudes are represented with reddish colors. The averaging kernels are only shown every 90 m in altitude for clarity.**

[Figure]

**Figure R.4.2: From left to right: averaging kernels of only-RL, only-MWR and MWR+RL. Each color corresponds to a different altitude: ground is represented by black, higher altitudes are represented with reddish colors. the *Ak are* calculated using a diagonal covariance matrix *Sa*. The averaging kernels are only shown every 90 m in altitude for clarity.**

**MINOR REMARKS:**

**L 42: Include also R. J. Sica and A. Haefele, "Retrieval of water vapor mixing ratio from a multiple channel Raman-scatter lidar using an optimal estimation method," Appl. Opt. 55, 763-777 (2016)**
Included in the manuscript.

**L 49: This is demanding but demonstrators exist. Include:**
**Dinoev, T., Simeonov, V., Arshinov, Y., Bobrovnikov, S., Ristori, P., Calpini, B., Parlange, M., and van den Bergh, H.: Raman Lidar for Meteorological Observations, RALMO − Part 1: Instrument description, Atmos. Meas. Tech., 6, 1329-1346, doi:10.5194/amt-6- 1329-2013, 2013.**

**Brocard, E., Philipona, R., Haefele, A., Romanens, G., Mueller, A., Ruffieux, D., Sime- onov, V., and Calpini, B.: Raman Lidar for Meteorological Observations, RALMO − Part 2: Validation of water vapor measurements, Atmos. Meas. Tech., 6, 1347-1358, doi:10.5194/amt-6-1347-2013, 2013.**
Included in the manuscript.

**L 141: Say explicitly how much the standard deviation is.**
Included in the manuscript.

**L 330: Something is wrong with "as to be expected 2".**
The number two was the reference to Fig. 2. It has now been corrected in the manuscript.

**L 350: The vertical resolution tends to infinity because the diagonal elements of the averaging kernels tend to zero. Include this explanation.**
Thanks for the clarification. The sentence has been included in the manuscript as follows:

"But outside this region, the vertical resolution for only-RL becomes infinite, because the diagonal elements of the averaging kernels tend to zero."

**L 354: Low resolution is bad, high resolution is good!**
Modified in the manuscript.

**L 391: It seems the panels of Fig. 5b are not in the right order. Reading the caption I understand 1.96 for combined, 0.84 for MWR and 0.96 for RL. The authors should also comment on the biases.**
The reviewer is right: there was a mistake in the figure caption, which has been now changed to:

"Figure 5. (a) Time series of IWV during the whole HOPE period from: the continuous GPS signal (black) and the one calculated from the joint retrieval, which is available only in clear sky cases (blue). Shaded areas represent the RL availability. (b) Scatterplot for the three cases: only Raman Lidar, only MWR and the joint retrieval (from left to right), against the GPS."

A sentence commenting the biases has also been included in line 391:
"While the only-MWR case presents a negative bias of ~0.5 kg/m2, the inclusion of the RL in the RL+MWR case, corrects this bias reducing it one order of magnitude. The combination of the two instruments and the only-MWR case present similar standard deviations, whereas the only-RL case presents a twice as large standard deviation in comparison to the other two cases. This results give us confidence that the developed OEM water vapor profiles are well constrained with respect to the integrated value. "

**L445: There is no a and b in Fig. 7.**
Corrected in the text.

***L 548: This does not sound right. It seems you scaled the variance by a factor of 4 and hence the standard deviation scales by a factor of 2. I expect in the RL region the a posteriori uncertainty if fully determined by the RL uncertainty.***

Here the standard deviation, and not the variance, was scaled by a factor of 4. The final error affecting the combined retrieval increases by a factor of 2-3, instead of 4, because of the stabilization by the prior.